There are amendments to this paper

# Inferring structural variant cancer cell fraction

Marek Cmero [1,2,3,4,5,920]*, Ke Yuan [6,7,920], Cheng Soon Ong [8,9,10], Jan Schröder[4], PCAWG Evolution and Heterogeneity Working Group, Niall M. Corcoran[1,2], Tony Papenfuss [4], Christopher M. Hovens[1,2], Florian Markowetz [7], Geoff Macintyre [3,7]* & PCAWG Consortium

We present SVclone, a computational method for inferring the cancer cell fraction of structural variant (SV) breakpoints from whole-genome sequencing data. SVclone accurately determines the variant allele frequencies of both SV breakends, then simultaneously estimates the cancer cell fraction and SV copy number. We assess performance using in silico mixtures of real samples, at known proportions, created from two clonal metastases from the same patient. We find that SVclone's performance is comparable to single-nucleotide variant-based methods, despite having an order of magnitude fewer data points. As part of the Pan-Cancer Analysis of Whole Genomes (PCAWG) consortium, which aggregated whole-genome sequencing data from 2658 cancers across 38 tumour types, we use SVclone to reveal a subset of liver, ovarian and pancreatic cancers with subclonally enriched copy-number neutral rearrangements that show decreased overall survival. SVclone enables improved characterisation of SV intra-tumour heterogeneity.

[1] Department of Surgery, Division of Urology, Royal Melbourne Hospital and University of Melbourne, Parkville, VIC 3050, Australia. [2] The Epworth Prostate Centre, Epworth Hospital, Richmond, VIC 3121, Australia. [3] Department of Computing and Information Systems, University of Melbourne, Parkville, VIC 3010, Australia. [4] Bioinformatics Division, The Walter and Eliza Hall Institute of Medical Research, Parkville, VIC, Australia. [5] Murdoch Children's Research Institute, Parkville, VIC 3052, Australia. [6] School of Computing Science, University of Glasgow, Sir Alwyn Williams Building, Glasgow G12 8RZ, UK. [7] Cancer Research UK Cambridge Institute, University of Cambridge, Cambridge CB2 0RE, UK. [8] Electrical and Electronic Engineering, University of Melbourne, Parkville, VIC 3010, Australia. [9] Machine Learning Research Group, Data61, Canberra, ACT 2601, Australia. [10] Research School of Computer Science, Australian National University, Canberra, ACT 2601, Australia. [920]These authors contributed equally: Marek Cmero, Ke Yuan. PCAWG Evolution and Heterogeneity Working Group authors and their affiliations appear at the end of the paper. PCAWG Consortium members and their affiliations appears online. *email: cmerom@unimelb.edu.au; geoff.macintyre@cruk.cam.ac.uk

The clonal theory of cancer evolution[1] posits that cancers arise from a single progenitor cell that has acquired mutations conferring selective advantage, resulting in the expansion of a genetically identical cell population or clone. As a cancer grows, a process akin to Darwinian species evolution emerges with subsequent genetically distinct populations arising from the founding clone via the continual acquisition of advantageous genomic aberrations. Consequently, tumours are likely to consist of a genetically heterogeneous combination of multiple cell populations, the extent of which has been revealed through the use of whole-genome sequencing[2,3]. As clones can respond differently to therapy[4], understanding this cellular diversity has important clinical implications[5].

The mutations belonging to each clone in a tumour can be interrogated using bulk whole-genome sequencing, with mutation detection subject to factors such as sequencing depth and quality, tumour cellularity and mutation copy number[6]. The expansion of each clone over the life of a tumour is encoded in the allele frequency of somatic mutations[7]. To characterise the clonal composition of a tumour, the variant allele frequency (VAF) must be converted to a cancer cell fraction (CCF), the fraction of cancer cells within which the variant is present. Events appearing in all cancer cells (CCF = 100%) are considered clonal and due to a pervasive expansion. Events appearing in a subset of cells (CCF < 100%) are considered subclonal and part of an ongoing expansion. Estimating the cancer cell fraction of events is challenging, as the observed variant allele frequency depends on the amount of normal cell admixture (purity) and local copy number.

Given these challenges, previous computational approaches for estimating CCF have focused on individual facets of this complexity, commonly limiting their view to single-nucleotide variants (SNVs)[8–13] or somatic copy-number aberrations (SCNAs)[14–16]. This has left the clonality of balanced rearrangements largely unexplored, despite their implication in oncogenic fusions[17] and subclonal translocations conferring drug-resistant phenotypes[18]. While SNV-based approaches have provided solutions to the problem of downstream inference of mutation CCF, they cannot be used for structural variant (SV) breakpoint data as: (i) no complete and robust methodology exists yet to calculate VAFs from SVs (Fan et al.[19] provides a limited framework that does not correct for DNA-gains or support all SV types), (ii) SVs themselves can cause copy-number changes (background copy numbers must therefore be inferred differently), (iii) SVs are composed of two ends, each with a potentially different VAF, and (iv) due to the relatively small number of data points (on average compared with SNVs), false-positive SVs greatly diminish clustering performance, hence a robust filtering methodology is required to consider only high-confidence SVs.

To address this gap, we present SVclone, an algorithmic approach that infers CCFs of SV breakpoints. It considers all types of large-scale structural variation (SV), including copy-number aberrant and copy-number neutral variation. The Pan-Cancer Analysis of Whole Genomes (PCAWG) Consortium has aggregated whole-genome sequencing data from 2658 cancers across 38 tumour types generated by the ICGC and TCGA projects. These sequencing data were re-analysed with standardised, high-accuracy pipelines to align to the human genome (reference build hs37d5) and identify germline variants and somatically acquired mutations, as described in[20]. Here we apply SVclone to these large-scale data to generate insight into patterns of clonality of structural variation across a large number of cancer types, and identify functionally important and clinically relevant observations.

## Results

**Algorithm overview.** The SVclone algorithm consists of five steps: annotate, count, filter, cluster and post-assign. A graphical representation of the SVclone pipeline can be found in Fig. 1a. Here we briefly summarise each step with detailed explanations appearing in the Methods section.

Annotate: SV calls are required as input into the annotate step (single-nucleotide resolution paired SV loci), and the corresponding whole-genome sequencing file in BAM format. The annotate step determines the read directionality of SVs and classifies the SV type.

Count: The count step estimates the supporting and normal (non-supporting) read counts and computes SV VAFs.

Filter: The filter step removes low-quality SVs and those with missing information, and, given copy-number calls, infers the background copy number for each break-end.

Cluster: The cluster step simultaneously estimates the mutated copy number of SVs, the number of clusters and their respective CCF means. Allele frequencies from both break-ends of each SV are used to perform inference.

Post assign: The post-assign step (re)assigns variants a most-likely mutated copy number and CCF, given the previously obtained clustering configuration.

**Estimation of SV allele frequency.** SV variant allele frequencies can be estimated in the same way as SNVs: the number of variant reads divided by the total number of reads observed at the SV breakpoint. The challenge for SVs is that many reads are split across the breakpoint making extracting accurate estimates for these read counts difficult. To explore how best to deal with this challenge we simulated reads from SVs with known allele frequency, at varying tumour purity. We then implemented an optimised approach for computing a VAF from these read counts (see Methods). The simulations revealed that the VAF estimates were accurate, independent of purity, except for duplications (Fig. 1c). Duplications showed an increased normal read count due to DNA gains showing no loss of normal DNA (Fig. 1b). To account for this bias, we introduced a scaling factor that incorporates tumour purity to calibrate the supporting read counts. This corrected for the bias and showed accurate estimation of the underlying VAF (Fig. 1c).

**In silico subclonal mixing of tumours for validation.** Recently, a number of efforts have been made to simulate datasets with known subclonal structure to assess the performance of algorithms that infer the CCF of mutations[21,22]. However, these have been limited to simulating SNVs and copy-number changes. To date, a gold standard dataset to test the performance SV cancer cell fraction inference does not exist. Therefore, we created a dataset of tumour samples with known SV subclonal structure. Rather than simulate SVs, we opted to mix two whole-genome sequenced samples from the same patient[23], in silico, at known subclonal proportions (Fig. 2a). By mixing tumour sequence data, we maintained many of the noise characteristics of real sequence data. Our samples consisted of a set of three-cluster mixtures with SV and SNVs subsampled with known clonal frequencies at 10% increments, as well as four and five-cluster mixtures created by subsampling odd and even chromosomes at different frequencies (Fig. 2a). The prostate cancer samples used to create the mixtures had no evidence of subclonality (Supplementary Fig. 2d from Hong et al.[23]), and had similar read coverage and tumour purity.

**Optimal cancer cell fraction versus ground truth.** Our in silico mixtures allowed us to explore some of the fundamental noise properties of CCF distributions. As the read counts supporting the SVs and SNVs in our mixed samples were subject to noise (approximately binomially distributed), we hypothesised that the resulting CCF estimates must also be noisy (approximately

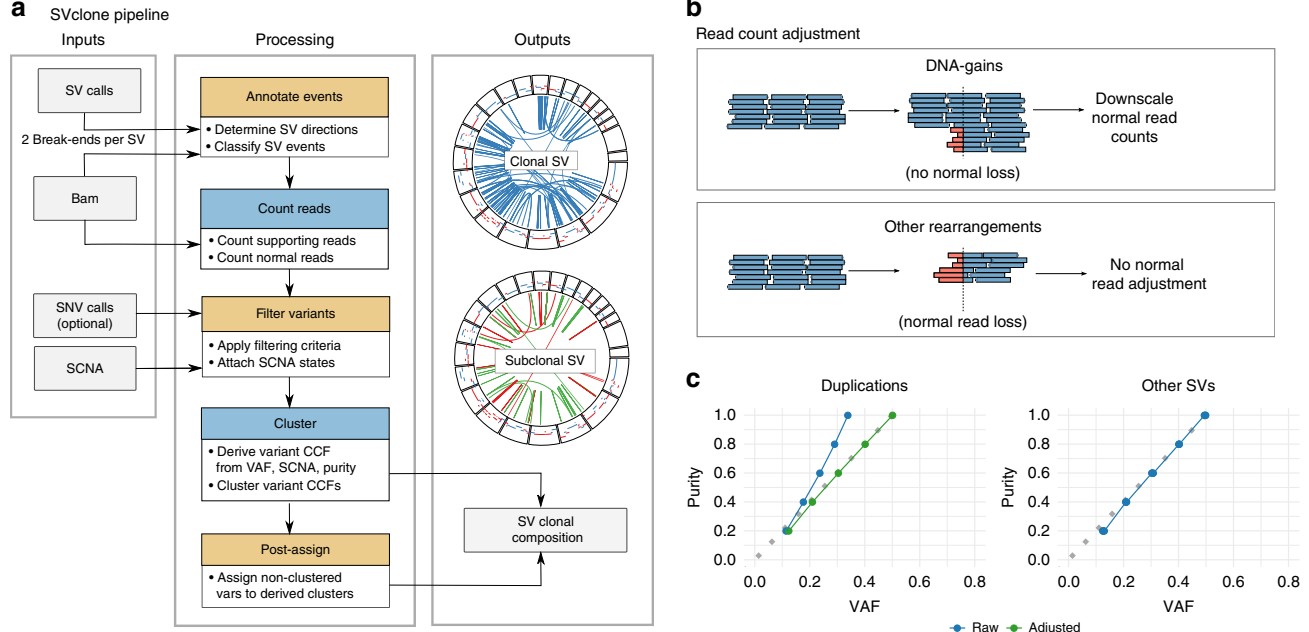

**Fig. 1 Pipeline schematic and VAF calculation adjustment. a** A flow-chart of the SVclone pipeline. **b** A schematic showing the adjustments required for DNA-gains (top) and all other rearrangements (bottom). From left to right, each segment shows an unaffected locus, the effect of the variant type on reads at the breakpoint, and the resulting adjustment strategy required to normalise the allele frequency. Red portions of the reads show soft-clips, i.e. the portion of the reads mapping to the other end of the breakpoint. **c** The effect of adjusting raw VAFs in duplications (left), and unadjusted VAFs for other SVs (right), at purity levels at 20–100% in 20% increments, where the expected VAF is half the purity level (dotted line).

normally distributed). To observe this, we estimated an 'optimal' CCF for each variant, which was calculated using the VAF and inferring multiplicity from the true mixture proportion (see Methods). This estimate allowed us to observe the optimal CCF distributions (Fig. 2c). Indeed, we observed that subclonal populations were approximately normally distributed, and those with similar CCFs had overlapping distributions (Fig. 2b).

These optimal CCF estimates also allowed us to explore any differences between SV and SNV CCF estimation. Overall, 234 high-confidence SVs and 9810 SNVs were called across both metastasis samples. The lower number of SVs resulted in the optimal CCF estimates having less clear CCF peaks than SNVs (Fig. 2b). These data highlight the difficulty in estimating CCFs for SVs as compared with SNVs. In addition, at the variant level, CCFs of SVs had a slightly higher mean error (ME) compared with SNVs (0.0461 vs. 0.015 across all cluster mixtures; per mixture results are shown in Fig. 2c).

Given the ground truth, these data also allowed us to determine the optimal per-variant cutoff for determining whether a variant was subclonal (Fig. 2d). We found that taking the max or mean CCF of both SV ends resulted in the highest AUC (~0.90), which was also approximately equal to the AUC obtained by classifying SNVs. The optimal CCF cutoffs for determining subclonality were 0.69 and 0.72 for SVs (using mean CCF) and SNVs respectively—to simplify this, we used a cutoff of 0.7 for both variant types for all downstream analyses.

**Performance assessment**. SVclone is chiefly designed to determine the CCF of SVs in a single, whole-genome sequenced tumour sample. Common downstream analyses of these data include analysing the number of subclonal populations in a sample[24] and observing which SVs are clonal or subclonal[25]. As such, we designed performance metrics to interrogate such variables including: cluster number error, mean cluster CCF error, mean variant CCF error, and sensitivity and specificity for calling

a variant subclonal. As one of the key features of any CCF inference algorithm is to estimate the number of chromosome copies of a variant (known as multiplicity), we also observed the mean multiplicity error.

To our knowledge, no other method for estimating SV CCF exists for direct comparison. Instead we opted to compare to two representative, state of the art methods for estimating the CCF of SNVs, PyClone[10], and copy number, Battenberg[15], from single samples. In addition, we also ran SVclone in SNV clustering mode, which uses Ccube's clustering model[26]. Performance is summarised in Fig. 3, and a breakdown of the performance under each measure can be found below.

Cluster number error: This metric indicates how effective the given clustering algorithms were at inferring the correct number of clusters. SVclone applied to the in silico mixtures was able to identify the correct number of clusters in 7 of 11 cases (Fig. 3). SVclone's SNV clustering found the correct number of clusters in 5/11 cases, compared with PyClone's 4/11, suggesting that SVs may have a slight advantage in identifying the correct number of underlying clusters.

Mean cluster CCF error: Mean cluster CCF error was generally higher in the SV data, with an average mean error of 0.0913, compared with 0.0412 and 0.0756 observed in the SNV data by SVclone and PyClone respectively. This is likely due to the variant number differences, as the comparatively larger number of SNVs is likely to lead to more accurate cluster CCF estimates.

Mean variant CCF error: Similarly, mean variant CCF error was slightly higher in the SV data than other methods. SV CCFs had an average mean error of 0.0873, compared with −0.034 for SVclone SNVs, −0.0213 for PyClone, and 0.0375 for Battenberg. Slightly higher error rates for SV CCFs are expected, given that the optimal (i.e. best obtainable given knowledge of cluster means) CCF mean errors averaged 0.0408 and 0.002 for SVs and SNVs respectively (Fig. 2c). Notably, while Battenberg performed on average better than SVclone in terms of mean variant CCF error for the three-cluster mixes, SVclone performed better on the

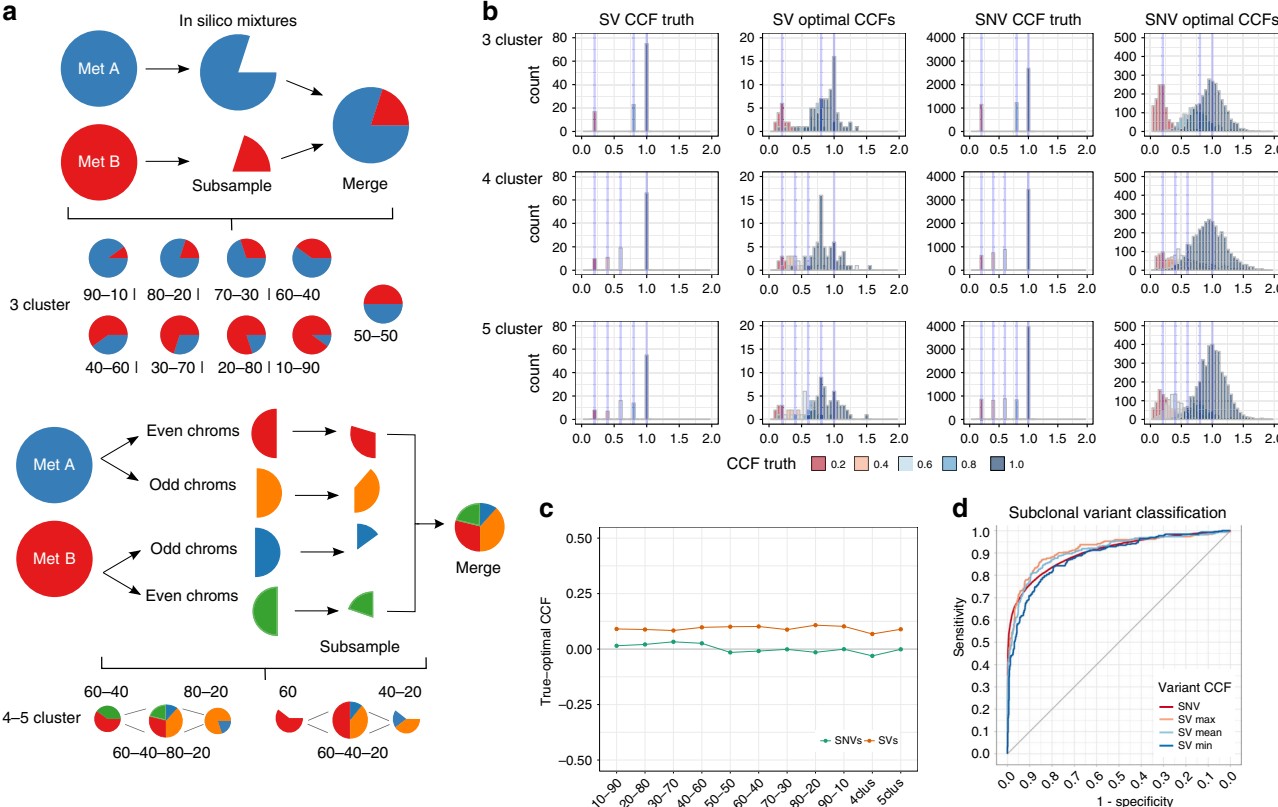

**Fig. 2 In silico mixing strategy and optimal CCF calculation metrics. a** A schematic illustrating the subsampling and merging process used to create in silico mixtures of real tumour samples. The top diagram shows the three-cluster in silico mixtures, created by mixing the two metastasis samples in different proportions. The bottom diagram shows the methodology for creating the four- and five-cluster mixtures, which separates each mixture sample into even and odd chromosomes, then subsamples these samples to create additional clusters. The resultant CCFs are based on the subsampling percentage of each odd or even chromosome sample, rather than the sample proportion (as in the three-cluster mixtures). **b** The CCF ground truth (based on sample membership) versus optimal SV and SNV results (based on transformed variant allele frequencies from the true cluster mean) for a representative three-cluster mixture and the four- and five-cluster mixtures. **c** Mean per-variant CCF error of optimal SNV and SV CCFs compared with the expected, ground truth CCF. **d** ROC curves for classifying variants as clonal or subclonal based on optimal variant CCFs.

four- and five-cluster mixtures, demonstrating SVclone's advantage in being able to consider >2 subclones. SVclone's SNV clustering and PyClone displayed similar mean error trends across the mixtures. Given the relatively smaller number of variants used in the clustering compared with SNVs, and the fewer data points used to infer fraction compared with SCNAs, SV CCF mean errors were in general comparable to other methodologies, with <0.05 absolute difference, on average, across the mixtures.

Sensitivity and specificity for calling a variant subclonal: SVclone's SV estimates demonstrated similar sensitivity to SNVs when classifying a variant as subclonal, with an average sensitivity of 0.670 (compared with an SNV sensitivity of 0.6643). The SVs had a lower specificity (0.8852 vs. 0.952 with SNVs). PyClone displayed a lower sensitivity, but higher specificity than the other methods at 0.577 and 0.9687 respectively. Battenberg had the highest average sensitivity and specificity (sens = 0.747, spec = 0.9175), which is expected given the number of data points (germline SNVs) used by Battenberg to infer each copy-number fraction.

Multiplicity error: Multiplicity error represents the difference in the multiplicity inferred from clustering, compared with the inferred multiplicity given the 'true' CCF cluster mean (as multiplicity cannot directly be observed). As PyClone averages across all possible multiplicities, and does not directly estimate multiplicity, we did not consider PyClone for this metric. Average multiplicity errors were −0.0391 for SVs and 0.1029 for SNVs.

The lower multiplicity error rate in SVs is likely due to the subclonal copy-number inference model (only SNVs with clonal copy numbers were considered), which allows for non-integer copy numbers. The mean multiplicity error for clonal SVs across the three-cluster mixtures was −0.1239, similar in absolute terms to the SNV multiplicity error (0.1029).

SVclone's comparable performance to SNV-based clustering indicates that clonal structure can be effectively reconstructed with high concordance and accuracy, despite the relative deficit in variant number. This means that the clonal structure of a tumour can be inferred from SNVs and SVs independently and their results compared. However, if it is assumed that the clonal populations in a sample share the same SNVs and SVs, we have also provided an option to cluster both SVs and SNVs using the same clustering framework. This is particularly powerful when considering model-based post-assignment. SNV CCF posterior can be integrated with SV read counts' likelihood to make assignment calls and vice versa (see Supplementary Fig. 1). By combining these data types overall performance can be increased.

Two of SVclone's unique design features also warranted further performance assessment: (1) SVclone incorporates background SCNA states from both breakpoint ends into its clustering model; and (2) SVclone clusters variants in clonal and subclonal copy-number regions. Here, we sought to quantify the advantages of both approaches over 'naive' approaches which considered only one breakpoint for each SV, or used only variants in clonal copy-number regions.

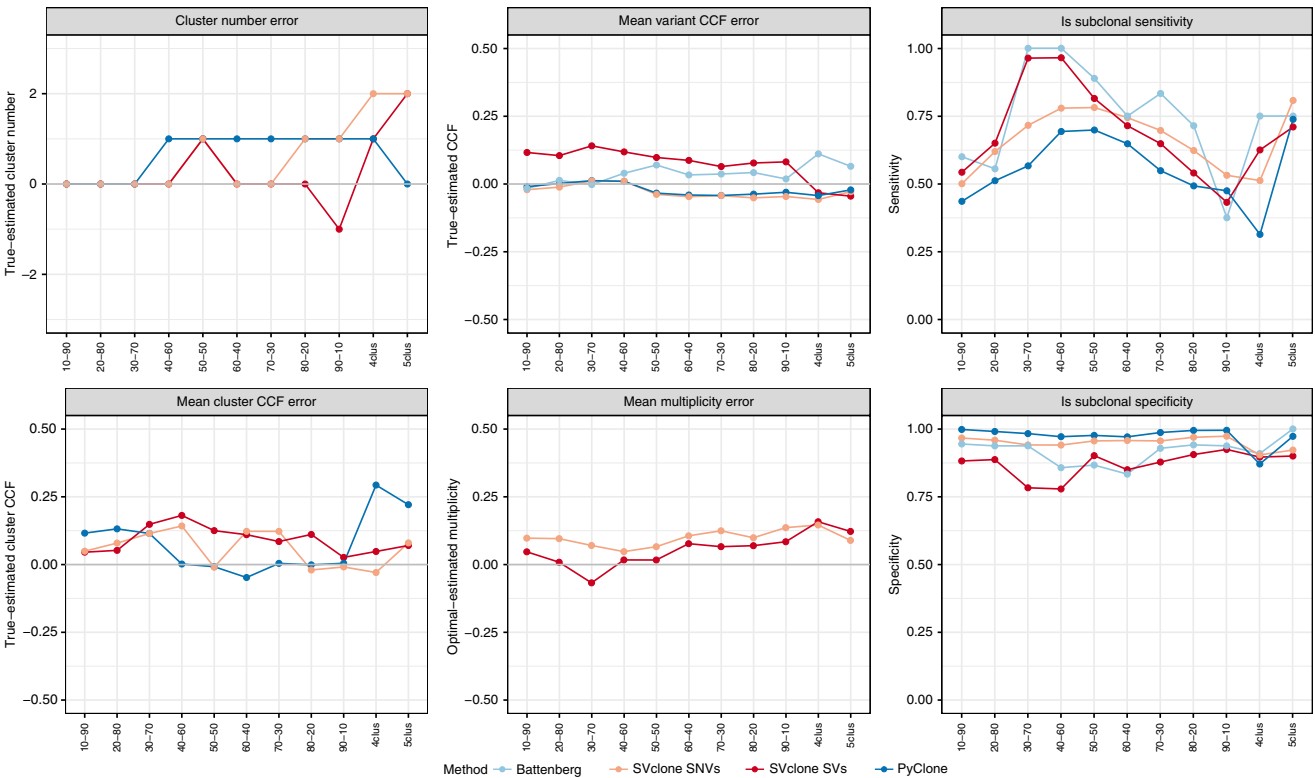

**Fig. 3 Clustering performance metrics versus existing methods.** Performance of SVclone's SV and SNV models, compared with Battenberg (SCNAs) and PyClone (SNVs) run on the in silico mixtures. The first column shows the cluster number error (three-inferred cluster number), and the mean CCF error, where true and inferred clusters are matched based on their order (see Methods). The second column shows the mean variant CCF and multiplicity error compared with the ground truth CCF. The third column shows the subclonal classification sensitivity and specificity using sample membership of the variant (i.e. a variant is classified as clonal if present in both samples of the mixture, and subclonal otherwise).

To compare the performance of SVclone's dual-end clustering model, to a single end, we ran the respective SV sides from the three-cluster in silico mixtures through SVclone's single-end (SNV) model, and compared the clustering performance to the dual-end model. Performance is summarised in Fig. 4. Figure 4 shows that dual-end model outperforms the single-end model across mean variant CCF error, mean multiplicity error, and mean cluster CCF error across almost all mixes. Only the cluster number of the 50–50 mix was incorrectly inferred, compared with the single-end model which was correct, However, we would expect only two clusters given the 50–50 mixture split and thus the dual-end model's result is likely more parsimonious with the data. Interestingly, the single-end model showed a higher subclonal classification sensitivity, but a lower specificity than the dual-end model. Given that this metric represents a sensitivity and specificity trade-off, we generated a ROC curve (Supplementary Fig. 2). Considering the AUC indicates that the dual-end model is preferable (AUC of 0.8234 vs. 0.8095 for the dual and single-end models respectively).

We further hypothesised that the dual-end model was more robust to SCNA noise. To investigate this point, we selected the 70–30 in silico mixture due to its low variant CCF error, and perturbed copy number in the following ways: (i) major allele copy number − 1 (CN − 1 for short), (ii) major allele copy number + 1 (CN + 1), and (iii) subclonal copy-number fraction +/− 0.3, where, 0.3 is added to the copy-number fraction if the resulting fraction is <0.9, otherwise we subtract 0.3 (see Methods for further details). We performed these experiments for the dual-end model, perturbing one side and both sides in separate runs. As expected, we found that the CN-perturbed runs showed slightly worse performance across the measured metrics compared with the

unperturbed runs (Supplementary Fig. 3). In general, variant-level metrics were more severely affected than cluster-level metrics. All perturbations performed similarly, with CN − 1 (experiment i) on both sides being the most affected scenario. Mean variant CCF was most significantly affected with a 0.27 error in the CN − 1 scenario on both sides (compared with 0.07 in the unperturbed model). Mean cluster CCF error was only mildly affected, but was also most significant for the CN − 1 on both sides scenario (0.16 vs. 0.11 ME in the unperturbed data). The CN − 1 experiments were the only ones that caused an error in the cluster number. Supplementary Fig. 4 shows the effects of the perturbation experiments on the single-end model versus the dual-end model (where only one side is perturbed). The dual-end model was more robust to perturbation across all metrics for all perturbations except for cluster number with the CN − 1 experiment (where one extra cluster was called), subclonal classification sensitivity in the CN − 1 experiment and a slightly worse mean multiplicity error in the CN + 1 scenario. Interestingly, mean cluster CCF error was still lower in the over-clustered case. Importantly, the mean variant CCF error and mean cluster CCF error were lower in all cases when considering the perturbed dual-end model versus the perturbed single-end. In summary, these data show that the dual-end model is more robust to copy-number noise than the single end. Copy-number addition errors were better tolerated than subtraction errors, and a mis-estimation of copy-number fraction resulted in errors somewhere between the two. However, mean cluster CCF error and cluster number were minimally affected, suggesting that poor CN estimation effects are largely restricted to errors in variant-level estimates.

Finally, we compared SVclone's performance using SVs in both clonal and subclonal copy-number regions, to clonal only.

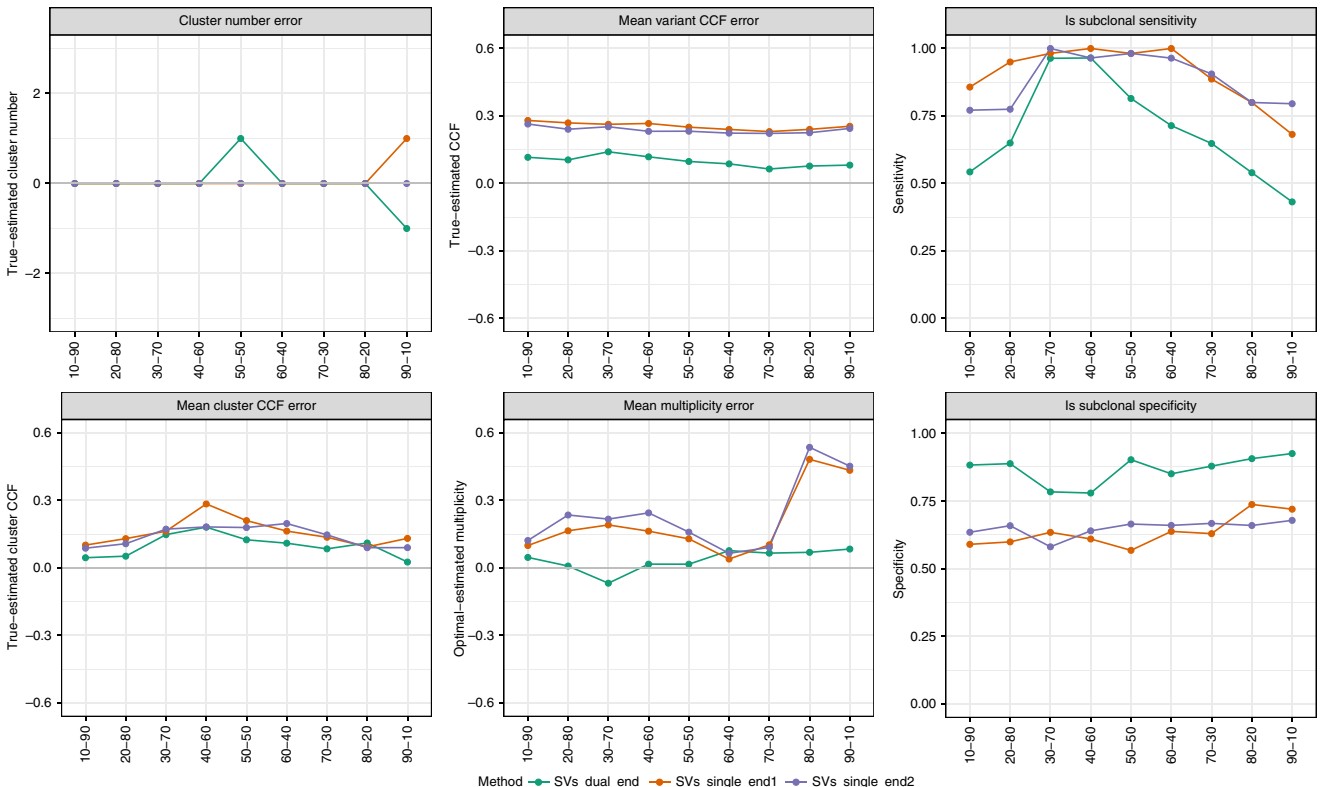

**Fig. 4 SV clustering performance for dual versus single-end models.** Performance of SVclone run on three-cluster in silico mixtures using either both breakends of an SV, or a single end. The first column shows the cluster number error (three-inferred cluster number), and the mean CCF error, where true and inferred clusters are matched based on their order (see Methods). The second column shows the mean variant CCF and multiplicity error compared with the ground truth CCF. The third column shows the subclonal classification sensitivity and specificity using sample membership of the variant (i.e. a variant is classified as clonal if present in both samples of the mixture, and subclonal otherwise).

Performance is summarised in Fig. 5. Utilising all available SVs improved the performance significantly across all metrics (apart from subclonal classification specificity) compared with clustering SVs with clonal background copy numbers states only.

**Clonality analysis of 1705 whole-genome sequenced tumours.** We applied SVclone to 1705 WGS samples from the pan-cancer analysis of whole genomes (PCAWG) project (dcc.icgc.org/pcawg)[20,27], clustering both SVs and SNVs separately. An analysis of the clonality of putative driver SV events can be found in Dentro, et al.[30] Here, we sought to observe any differences in the clonal structure of SVs compared with SNVs. Downstream analysis was performed on 23 tumour types showing ≥20 samples, with >10 SVs, and >10 SNVs, and sufficient power to detect subclonality (total $n = 1169$, see Methods).

A comparison of the fraction of subclonal SVs versus SNVs showed different patterns across tumour types (Fig. 6a). Tumour types showing a greater proportion of subclonal SVs versus SNVs included 100% of lung squamous cell carcinomas, and 92% of both colorectal adenomas and ovarian adenocarcinomas. In contrast, 23% of biliary adenocarcinomas had a greater proportion of subclonal SNVs versus SVs (Supplementary Table 1). Some cancers also contained subsets of samples with distinct patterns of clonality, for instance, liver cancers contained a cluster of 19 samples with high SV subclonality (≥50%) and low SNV subclonality (<30%).

One unique feature of SVclone is that it determines the clonality of copy-number neutral rearrangements (inversions and inter-chromosomal translocations). We applied a test for

enrichment of subclonal copy-number neutral rearrangements across the PCAWG cohort. A total of 177 samples across 28 cancer types exhibited a subclonal copy-number neutral rearrangement (SCNR) pattern (e.g. Fig. 6c–f, see Supplementary Fig. 5 for the distribution of the pattern across histologies), with ovarian ($n = 29$, 25.7% of total ovarian), liver hepatocellular carcinoma ($n = 26$, 10.4% of liver samples) and pancreatic cancers ($n = 18$, 7.5% of total pancreatic) overrepresented in this set.

To test for potential clinical relevance of this SCNR pattern, we compared the overall survival of SCNR cases ($n = 177$), with high SV heterogeneity cases ($n = 650$), and all remaining cases ($n = 447$) for which overall survival was recorded, stratified on age, tumour histological subtype, and number of SVs. These groups showed significantly different survival probabilities ($p = 0.006$, likelihood-ratio test), with median survival times of 1236, 1470 and 2907 days, respectively (Fig. 6b). This resulted in a hazard ratio of 1.930 for SCNR cases, significantly higher compared with the baseline cohort ($p = 0.0014$, Z-test). In contrast, the high SV heterogeneity cases had a hazard ratio of 1.302 ($p = 0.084$, Z-test). Given the high number of ovarian samples within the SCNR cohort, we also considered whether fold-back inversions (FBI) were enriched, as they have been previously associated with poor prognosis[28]. We found no evidence for enrichment of FBIs (see Supplementary Fig. 6 and methods for further details), suggesting that the SCNR genotype might arise from an independent mechanism.

To test if these SCNR events were the result of a single complex rearrangement event (such as chromothripsis), or were simply a set of unrelated rearrangements, we looked for clustered events, and where possible, attempted to walk the derivative chromosome

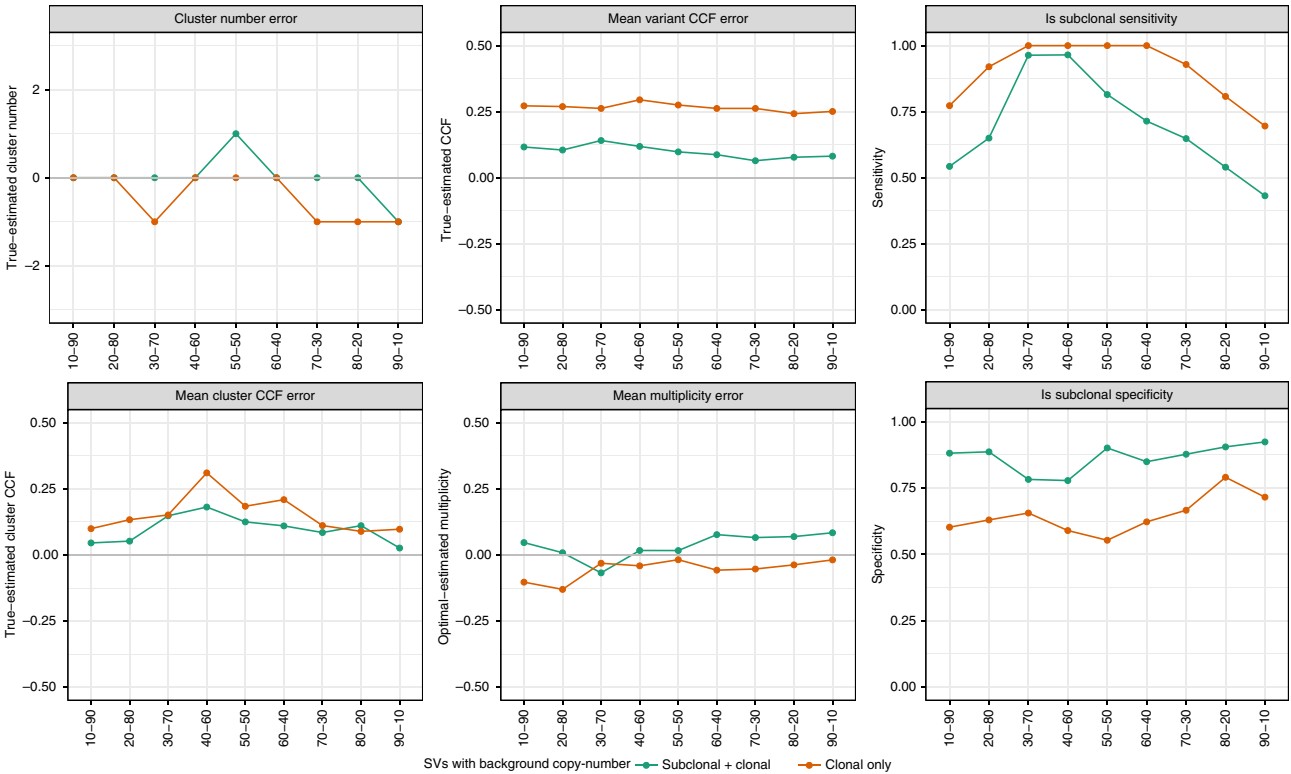

**Fig. 5 SV clustering performance incorporating background subclonal copy-number states.** Performance of SVclone run across the three-cluster in silico mixtures using either clonal background copy-number states, or clonal plus subclonal states. The first column shows the cluster number error (three-inferred cluster number), and the mean CCF error, where true and inferred clusters are matched based on their order (see Methods). The second column shows the mean variant CCF and multiplicity error compared with the ground truth CCF. The third column shows the subclonal classification sensitivity and specificity using sample membership of the variant (i.e. a variant is classified as clonal if present in both samples of the mixture, and subclonal otherwise).

(i.e. Korbel's[29] fourth statistical criterion for chromothripsis, indicating that the fragments of a given chromosome form a 'walkable' chain of segments through consistent orientation). SCNR events were ~26% more likely to be part of a complex event, compared with background (39% vs. 31% in absolute terms; $p = 0.002$, two-sided $t$ test on proportion of linked SVs between SCNR and other samples). Overall, 50% of these clustered SCNR events were linked by at least one interchromosomal translocation, compared with only 22% of other samples (see Supplementary Table 2), suggesting these events can span multiple chromosomes. We found a slight increase between the fraction of chromosomes that could be walked between SCNR (2.7%) and other samples (2.1%), however this was not significant ($p = 0.4029$, two-sided $t$ test). Overall, these data suggest that subclonal events present in SCNR samples are likely enriched in complex, interrelated rearrangements.

To provide some insights into the aetiology of the SCNR genotype, we looked for an enrichment of SNVs/INDELs in known cancer drivers, which may cause the SCNR genotype. Specifically, we considered clonal (CCF > 0.7) mutations as they may reveal predisposing drivers to an SCNR genotype. We found an enrichment of *TP53* mutations (40.11% of SCNR samples) compared with background (14.54% of other samples; FDR < 0.0001, hypergeometric test). The enrichment of *TP53* is consistent with the reported link between TP53 mutations and complex rearrangements[30]. However, an enrichment of *TP53* SNVs/INDELs was also observed in the high SV heterogeneity cohort (36.77% of high SV heterogeneity samples), along with KRAS and CTNNB1 (all FDR < 0.001), suggesting that *TP53* may be necessary but not sufficient for an SCNR genotype.

Finally, to determine whether the enrichment of subclonal neutral SVs within SCNR samples harboured functional consequences, we identified all driver genes with candidate bi-allelic hits involving an SCNR. We considered a candidate bi-allelic hit as two separate mutation events affecting the same gene (copy-number loss, an SV within the gene body and/or an SNV/INDEL). We found that 62.15% of SCNR samples had at least one subclonal balanced rearrangement affecting a driver gene that was also affected by another mutation, almost double the rate found in the high SV heterogeneity cohort (32.15%) (Supplementary Table 3). This indicated that functionally-relevant consequences of the SCNR genotype are likely.

## Discussion

Here we have presented an integrated method for inferring the cancer cell fraction of structural variation breakpoints, and have demonstrated the importance of considering the clonality of neutral rearrangements. In cancers where copy-number neutral rearrangements are common, a significant portion of the clonal landscape has remained, until now, unexplored.

Despite the successful applications of SVclone demonstrated here, it is important to consider some of its limitations. In this work, our clustering model considers all SVs as independent events despite the fact that in some cases these SVs may be part of the same complex rearrangement. Complex rearrangements are not identified by SVclone's classification framework, however, users may specify their own types, if known. As more sophisticated methods for classifying complex SV events become available, this could be integrated into the algorithm framework. Another limitation to consider is that all CCF clustering-based methods are affected by the power to detect

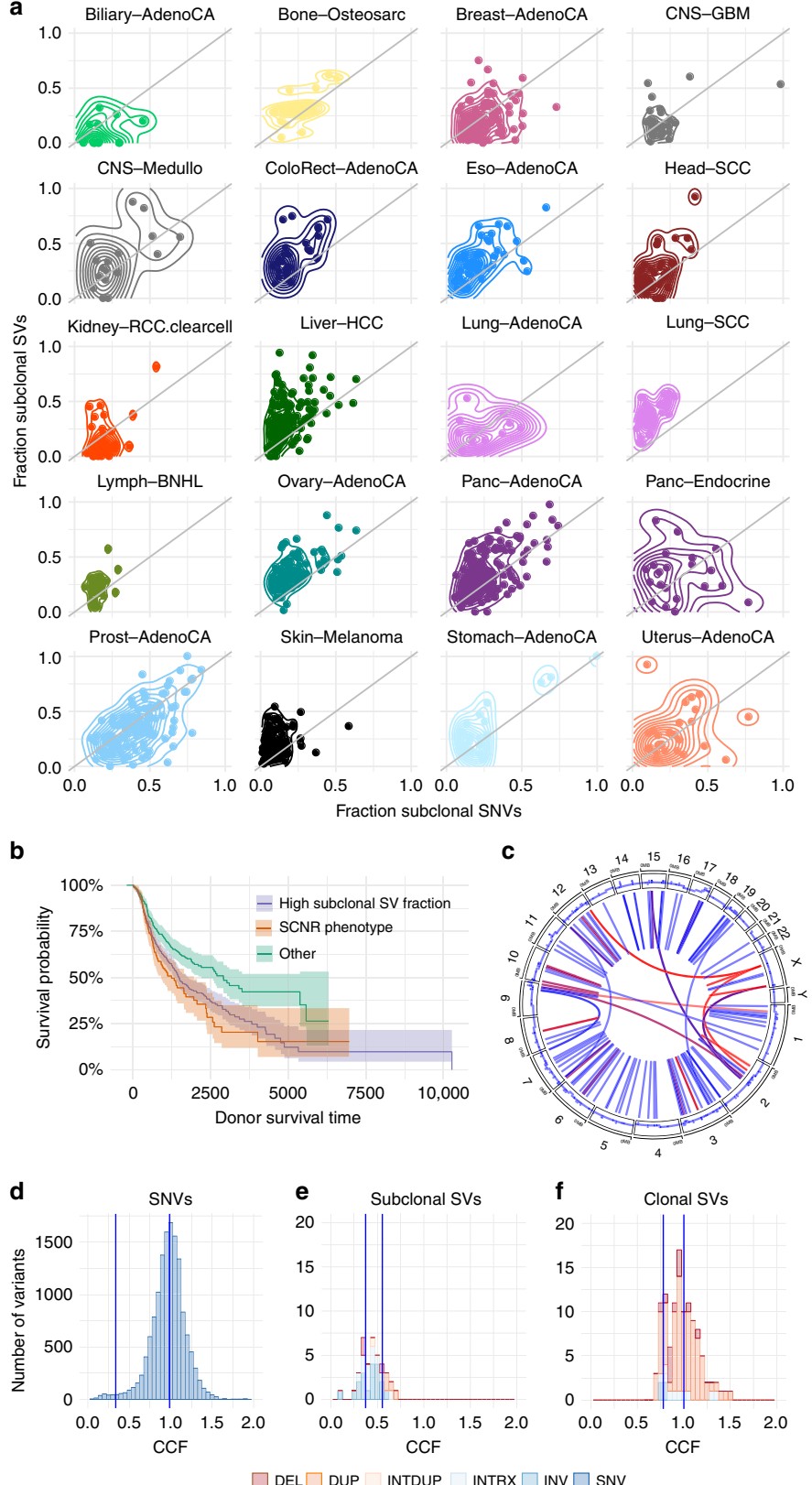

variants and accurately estimate their VAFs. We present an extensive analysis investigating the effects of tumour purity, coverage and copy number (for SNVs) on the power to detect clones and subclonal mutations in Dentro et al.[25], which is also applicable to SVs.

Inferring the evolutionary history of SVs from whole-genome sequence data is a challenging problem. One of the key goals in the field is to derive a clone tree that depicts the acquisition of SVs over time and their relationship to clonal expansions during tumour evolution. To achieve this, a number of key variables

**Fig. 6 Application of SVclone to PCAWG cohort. a** A 2D density plot of the fraction of subclonal SVs versus SNVs for PCAWG samples ($n = 1169$) (a variant under 0.7 CCF was considered subclonal). **b** Survival curves representing patients divided into those with a SCNR pattern, those with high subclonal SV fraction, or neither. **c** A circos plots for an example SCNR pattern tumour (Liver Hepatocellular carcinoma, tumour WGS aliquot 2bff30d5-be79-4686-8164-7a7d9619d3c0). The outside track represents the copy number across the genome and the inner lines indicate SVs. Blue lines represent clonal SVs and red lines represent subclonal SVs. **d** A CCF histogram of sample 2bff30d5-be79-4686-8164-7a7d9619d3c0's SNVs. **e** A CCF histogram of 2bff30d5-be79-4686-8164-7a7d9619d3c0's subclonal SV's colour coded by SV category. **f** A CCF histogram of 2bff30d5-be79-4686-8164-7a7d9619d3c0's clonal SVs.

must be inferred from the data: variant allele frequencies of SV breakpoints; number of DNA copies harbouring SV breakpoints (also known as multiplicity), the cancer cell fraction of SVs, cancer cell fraction of clones, and a clone phylogeny. No one method exists that can simultaneously infer all variables, but rather existing methods tackle subsets: Fan et al.: VAF[31], WEAVER: VAF + clonal multiplicity[32], TUSV: subclonal multiplicity + clone CCF + phylogeny + (additionally) clone copy number[33], Meltos: VAF + phylogeny[34], and SVclone: VAF + subclonal multiplicity + approximate clone CCF + SV CCF. At present these methods need to be combined to achieve a more complete picture of the evolution of SVs (e.g. WEAVER + TUSV[33] or SVclone + Meltos[34]). Thus, there remains an opportunity for future development of an algorithm that can simultaneously infer all variables.

Inferring the evolution of all variant classes, including SVs, SNVs, SCNAs, indels, and their respective clonality will ultimately be required to gain a more complete picture of the tumour heterogeneity landscape. We have presented an integrated software package for modelling the cancer cell fraction of structural variation breakpoints using single sample whole-genome sequencing data and have demonstrated its application by identifying patterns of subclonal variation. This software enables further exploration and quantification of tumour heterogeneity, and moves us closer to an integrated approach to modelling tumour heterogeneity.

## Methods

**Data input**. The SVclone algorithm requires, at a minimum, a list of SV breakpoints and associated tumour BAM file. SV breakpoints can be provided as a VCF or as a tab-delimited file of paired single-nucleotide resolution break-ends. Using an SV caller with directionality of each break-end is recommended. The Socrates[24] output format is natively supported and allows additional filtering by repeat type and average MAPQ. An associated paired-end, indexed whole-genome sequencing BAM file is required. In the filter step, copy-number information can be added in Battenberg[15], ASCAT[35] or PCAWG consensus copy-number formats to aid in correcting VAFs. SNV input is also supported in multiple VCF formats (sanger, mutect, mutect call-stats and PCAWG consensus). Further details of input formats can be found in the repository README file.

**SV annotation**. To accurately calculate variant allele frequency (VAF) of structural variants, the following information is required: (i) the single-nucleotide location of loci comprising each breakpoint; (ii) the direction in which the break faces, i.e. whether the breakpoint is on the left ($-$) or the right side ($+$) of a locus that connects to the distant locus; and (iii) the classification of the SV. SV directionality affects read counting, as only reads on one side of each break-end will correspond to a specific breakpoint.

SVclone incorporates basic methodology to infer the breakpoint direction (ii) and classification (iii) of the SV, however, we recommend using the information provided by the SV caller if it is available. SVclone will infer the directionality of each break-end by determining which side of the break-end has soft-clipped reads. If SVclone finds evidence of soft-clips lying on both sides of the break-end (i.e. at least 10% of soft-clipped reads support the opposite directionality), we consider the directionality for this break-end as mixed (i.e. multiple break-end pairs are involved for this event). If only one break-end of a pair has mixed directionality, the SV will be split into two events, one where the mixed-evidence locus is ($-$), and the other end is ($+$). If both ends have mixed directionality, we attempt to resolve this by searching the SV input for other SV events matching the SV break-ends, considering the following scenarios (see Supplementary Table 4 for a summary). We denote each SV as $j = 1..J$; $i \in [l, u]$ where $l =$ lower break-end locus and $u =$ upper break-end locus, $l < u$ if the chromosome is the same, or the lower of the chromosomes for inter-chromosomal translocations.

**SV directionality inference**. Directionality is determined for each SV as follows: (i) neither $l$ nor $u$ matches any other event: the SV breakpoint is considered to be ($-$, $-$), and a new SV breakpoint is created with directionality ($+$, $+$); (ii) both $l$ and $u$ match (within a threshold): we consider one pair's directionality as ($-$, $-$) and the other pair's as ($+$, $+$); (iii) two matching breakpoints are found, each break-end matching one locus of each partner only: if the positional rankings of the three SV breakpoints (on a single chromosome) are $[(1, 2), (2, 3), (1, 3)]$, we consider this a translocation event, and assign the directions of $[(+, -), (+, -), (-, +)]$; (iv) more than two matching breakpoints are found: the SV breakpoint is considered a complex event, and is discarded at the count step.

The directionality inference does not utilise local realignment of reads. The functionality is not intended to provide comprehensive and robust annotation. We recommend that directionality be inferred from the SV caller of choice.

**SV classification**. After resolving directionality, we employ a decision-tree based approach to classify SV events into categories if this information is unavailable from the SV caller (see Supplementary Fig. 7). We consider six simplified categories of rearrangements: inversions, deletions, tandem duplications, interspersed duplications and intra- and inter-chromosomal translocations. Inversions refer to a flipping of a segment of DNA, where the head of one segment joins the tail of another at both ends. Deletions are considered a loss of DNA at a locus where the flanking non-deleted segments join directly, without the intervening deleted sequence. Duplications are split into two categories: tandem and interspersed. The former category consists of a duplication joining tail to head immediately one after another. In the latter case, the duplication may be interspersed anywhere within the same chromosome. An intra-chromosomal translocation is similar to interspersed duplications, except that the original mobile element is deleted rather than retained. Inter-chromosomal translocations are defined as any joining event involving different chromosomes.

The classification heuristics are shown in Supplementary Fig. 8 and are summarised as: (i) inversion (INV): ($l$, $u$) directionality matches, i.e. ($+$, $+$) or ($-$, $-$), and there are 1 or 2 breakpoints corresponding to the inversion event; (ii) deletion (DEL): ($l$, $u$) directionality is ($+$, $-$), where $l < u$; (iii) tandem duplication (DUP) - breakpoint directionality is ($-$, $+$), where $l < u$; (iv) interspersed duplication (INTDUP): requires two breakpoints, $(l, u)_1$ and $(l, u)_2$ where $l_1 \approx l_2$ (within 100 bp) and $u_1 \neq u_2$ (one breakpoint has a tandem duplication signature and the other a deletion signature, i.e. $(l, u)_1 = (-, +)$ and $(l, u)_2 = (+, -)$); (v) intra-chromosomal translocation (TRX): the same as an interspersed duplications, except with the presence of a third breakpoint $(l, u)_3$, classified as a deletion that spans the mobile element: $l_3 \approx u_2$, $u_3 \approx u_1$ and $(l, u)_3 = (+, -)$. To successfully classify a translocation, the deletion ends must be within 6 bp (by default) of both ends of the mobile element; and (vi) inter-chromosomal translocation (INTRX) - the only criteria is that the chromosomes of $l$ and $u$ do no match, no directionality is considered.

**Read counting**. We consider three types of reads that cross the respective break-ends ($l$, $u$) (within 6 bp):

$s_i = s_l + s_u$: supporting split reads at $l$ and $u$ respectively. These are variant reads (supporting the break) where one of the read-pairs lies across the break-end by a specified number of base-pairs, which must be greater than the soft-clip threshold (10 by default for 100 bp reads).

$c_j$: supporting discordant (spanning) reads, i.e. reads that span across the ($l$, $u$) breakpoint, where each read of the pair lies on one side of the break, effectively spanning the breakpoint (see Supplementary Fig. 9). The insert distance is calculated by both reads' distance from their respective breakpoint at both ends. One of the reads may also be soft-clipped at the breakpoint, and still be counted as a supporting discordant read (these reads are counted under the spanning read category). In addition, the read orientation of both reads is also checked to ensure both reads are oriented towards the break (this is always the case for a true spanning read supporting the breakpoint).

($o_l$, $o_u$): normal read counts at $l$ and $u$ respectively. Either the read or the insert between the reads must lie across the breakpoint locus. These are reads derived from alleles *not supporting* the breakpoint. The outside ends of each read pair must overlap the breakpoint boundary by at least the specified base-pairs (10 by default for 100 bp reads) to be counted. Reads must not be soft-clipped above a small threshold (6 bp by default).

**Supporting read calculation.** Supporting reads are only counted if reads match the specified break-end directionality; this avoids double-counting of reads for events where reads are present at both sides of the breakpoint, such as inversions and translocations (these events consist of ≥2 breakpoints per event). All reads that are counted towards the supporting or normal read totals must have an insert size (fragment size) $< \mu_{ins} + (3 \cdot \sigma_{ins})$, where $\mu_{ins} =$ the mean of the insert size and $\sigma_{ins} =$ the standard deviation of the insert size. (The insert size for supporting spanning reads is considered the adjusted insert size for this criterion.) This is a quality-checking measure to ensure only high-confidence reads are counted. We consider both spanning and split reads together as the total supporting read count: $b_{i,j} = s_{i,j} + c_{i,j}$.

SV breakpoints where at any break-end the average read depth exceeds $\lambda \cdot maxn_j$ are considered high depth regions and are ignored, where $\lambda$ is the expected number of reads per locus and $maxn_j$ is the maximum expected copy-number value (coverage and maximum expected copy number can be defined by the user). These breakpoints are likely caused by repetitive regions, rather than true copy-number amplifications, and are not suitable for inference of clonal composition. Bed filtering has been incorporated to automatically ignore breaks falling within specified regions (to accommodate blacklists such as DAC—www.encodeproject.org/annotations/ENCSR636HFF/).

In order to determine whether micro-homology was likely to play a large role in the read counting process, we analysed the distribution of breaks containing micro-homologies across the PCAWG samples used in the paper analysis (using PCAWG's consensus SVs v1.6). We found that the mean and median micro-homology lengths were 1 and 2.4 respectively. Micro-homologies ≤ 6 bp in length are handled by the variable threshold used by the read counting step. We found that 6.17% of SVs had micro-homologies greater than 6 bp and <1% of SVs had micro-homologies greater than 20 bp. Given the minority of SVs affected, handling of longer micro-homologies is outside the scope of this work, and such SVs should be filtered out.

**Non-supporting read calculation.** For each SV, normal reads are counted at the break-ends resulting in two normal read count totals ($o_l$, $o_u$). In the case where the SV results in a gain of DNA (interspersed and tandem duplications), the normal read count must be adjusted. We consider the SV classification $\kappa_j$ for an SV $j$, where $\kappa_j \in$ {DEL, DUP, INTDUP, INV, TRX, INTRX}(respectively: deletions, duplications, interspersed duplications, inversions, translocations and inter-chromosomal translocations). We define two subsets $\kappa_{gain} =$ {DUP, INTDUP} where normal reads at the variant population's break-ends are unaffected at the variant allele, and $\kappa_{non-gain} =$ {DEL, INV, TRX} where the normal reads at the variant population's breakends are replaced by supporting reads. We compute an adjustment factor,

$$AF_{norm} = 1 - \left(\frac{t}{n_p}\right),  \quad (1)$$

where $t$ is the tumour content and $n_p$ the tumour ploidy. The normal read counts of all DNA-gain events are then multiplied by this adjustment factor ($o_{i,j} = o_{i,j} \cdot AF_{norm}$ if $\kappa_j \in \kappa_{gain}$), while events that are not DNA-gains remain unadjusted.

**Anomalous reads.** Reads that cross the SV boundary but do not meet the requirements for split, spanning or normal reads are considered anomalous and do not contribute to read counts. Reads can be considered anomalous for numerous reasons: (i) the insert distance is greater than $\mu_{ins} + (3 \cdot \sigma_{ins})$, (ii) discordant reads do not face the break, (iii) the read is soft-clipped at both ends, (iv) the read is soft-clipped but is either not in the vicinity of the breakpoint, boundary or the soft-clip is below the threshold, or (v) the reads support the break in the opposite direction, but have not been called by the SV calling algorithm. To investigate points i-iv, we investigated anomalous reads in the 100% purity deletion simulations, and flagged an average of 8.74 anomalous reads per breakpoint per 246.18 considered (3.57%) from the extracted regions around both break-ends of a breakpoint (these reads are proximal to the breakpoint and may not directly cross it). Upon manual inspection, we found that anomalous reads largely fell in the (iv) category, i.e. insufficiently long soft-clips or the reads genuinely did not cross the breakpoint boundary. Manual analysis uncovered no consistent under-counting of supporting reads.

**Filtering variants.** While tumours may contain several thousand unique mutations, typically SVs number in the dozens to low-hundreds (for instance, in breast cancers[36]). With typically 10-fold fewer variants, each variant utilised in clustering has a higher influence on the clustering results. A conservative approach to filtering is therefore required to minimise noise propagated through variants with spurious read counts. The following filtering criteria have been implemented, with default values, to provide a baseline for minimising noise. These variants may be adjusted in cases by the user to tailor their noise thresholds to the samples under consideration. We filter on the following criteria:

**Germline variants.** The output from the count step for the corresponding patient's germline sample can be supplied to filter out any events where there is at least one supporting read in the germline for breakpoints that are considered the same event (both break-ends match directionality and are within 6 bp of each other).

**SV size.** If a breakpoint is on the same chromosome, SV size ($u - l$) must be larger than the fragment size (by default) as otherwise supporting and normal reads may be difficult to distinguish. This criterion is only considered for intra-chromosomal events.

**Minimum support.** The SV breakpoint must have at least one split and one spanning read supporting the break ($s_{i,j} \geq 1$, $c_{i,j} \geq 1$). Custom minimum values can be specified.

**Minimum depth.** The minimum supporting + normal reads must be greater than the minimum depth for each break-end: ($b_{i,j} + o_l$) > $b_{min}$ and ($b_{i,j} + o_u$) > $b_{min}$. (Default $b_{min} = 2$).

**Copy-number state.** If copy-number input is provided, either $l$ or $u$ must have a valid copy-number state for each variant. The major + minor copy numbers must be at least 1 for a state to be considered valid.

Optionally, in some instances it may be appropriate to filter on several further criteria:

**Copy-number neutral regions.** Filters out variants with copy-number states that are not 1, 1 for major, minor alleles. Used if copy-number calls are unreliable and sufficient regions of neutral copy-number exist.

**Subclonal copy-number regions.** This filter may be invoked to remove any variants with subclonal copy-number states. This reduces the copy-number search space, which is useful for clustering high numbers of variants.

**Assigning background copy-number states.** Allele-specific copy-number variation can be supplied as input to SVclone in order to attach copy-number states to break-ends. We assign the estimated copy-number state that occurred before the SV occurred. For intra-chromosomal SVs, this involves obtaining the copy-number state upstream of the lower break-end and downstream of the upper break-end. For inter-chromosomal translocations, we obtain the copy number in the opposite direction of the break-end directionality. See Supplementary Fig. 10 for a conceptual schematic and Supplementary Table 5 for the mathematical representation.

Battenberg[15] output format is preferential to capture subclonal CNAs, however, ASCAT[35] is also supported. If no CNA information is supplied, the algorithm assumes that the total tumour copy number ($n_{tot_t}$) matches the normal copy number ($n_{tot_n}$), with no subclonality. For robustness of the algorithm results, it is recommended that copy-number information be supplied if available. If Battenberg input is defined, the first solution set of segmentations in the input is considered. We define the total copy number as the sum of each clone's copy number, weighted by the clonal fraction:

$$n_{tot_t,i,j} = \sum_{r=1}^{2} \rho_{r,i,j} n_{tot_t,r,i,j},  \quad (2)$$

where $n_{tot_t,r,i,j}$ and $\rho_{r,i,j}$ are the total copy number and copy-number fraction per (copy number) clone $r \in 1, 2$.

**Clustering.** The clustering step of SVclone simultaneously computes SV CCFs and clusters SVs of similar CCF, based on purity, ploidy and copy-number status of the normal, reference and tumour populations. SVclone uses a bespoke clustering algorithm that takes read counts and copy-number states at both break-ends of the same SV as input, and utilises a Bayesian mixture model, implemented using variational inference, to approximate posterior distributions for unknown parameters. The algorithm determines the number of clusters automatically and infers average CCF per cluster, as well as the multiplicity of each variant (the number of mutated chromosomal copy). The model extends our previous method, Ccube[26], for estimating and cluster CCFs for SNVs by allowing it to deal with additional read and copy-number profiles from the two break-ends. This is achieved by assigning the two break-ends of an SV to the same CCF cluster. Below is a detailed description of our clustering method.

**Read distribution.** Let $i \in 1, 2$ and $j \in 1, 2, ..., J$ be the indexes of break-ends and breakpoints respectively. We assume the supporting read counts from both breakpoints are independently distributed following two different Binomial distributions. The joint probability mass function of the supporting read counts is the following:

$$p\left(\boldsymbol{b}_j | \boldsymbol{d}_j, \boldsymbol{f}_j\right) = \prod_{i=1}^{2} Binomial\left(b_{i,j} | d_{i,j}, f_{i,j}\right),  \quad (3)$$

where $b_{i,j}$, $d_{i,j}$, and $f_{i,j}$ denote the number of supporting reads, the number of normal reads, and the probability of observing one support read. The bold font variable are collections of these across both breakpoints, $\boldsymbol{b}_j = [b_{1,j}, b_{2,j}]$, $\boldsymbol{d}_j = [d_{1,j}, d_{2,j}]$, and $\boldsymbol{f}_j = [f_{1,j}, f_{2,j}]$.

We model the probability of sampling a variant read given variant locus $j$ at break-end $i$ as coming from a binomial distribution with trials $d$ (read depth $b_j + o$) and probability $f_{i,j,k}$:

$$b_{i,j}|d_{i,j}, f_{i,j,k} \sim Binomial\left(b_{i,j}|d_{i,j}, f_{i,j,k}\right), \qquad (4)$$

where $b_{i,j} = s_{i,j} + c_{i,j}$, ($s_{i,j}$ is the number of split reads and $c_{i,j}$ the number of spanning reads supporting the break). We assume the two breakpoints are conditionally independent of each other given the same CCF. In order to calculate $f_{i,j,k}$ we require the tumour purity estimate $t$ and copy-number information:

$$f_{i,j,k} = w_{i,j}\phi_k + \epsilon, \text{ with } w_{i,j} = \frac{t(m_{i,j}(1-\epsilon) - n_{tot_n,i,j}\epsilon)}{(1-t)n_{tot_n,i,j} + t n_{tot_t,i,j}}, \qquad (5)$$

where $n_{tot_n,i,j}$ and $n_{tot_t,i,j}$ are the total copy number of the normal and tumour population respectively and $\epsilon$ is the sequencing error constant. $\phi_k, k \in 1, ..., K$ represents the unknown CCF, and is indexed by $k$, representing the $k$th cluster. The other unknown parameter is $m_{i,j}$, the number of mutated chromosomal copies, also known as the multiplicity of the variant. See below for how these are inferred.

To test the appropriateness of the binomial distribution for SV allele frequencies, we studied the distribution of clonal SV VAFs from the two samples used in the in silico mixtures (see below for more details). A likelihood-ratio test was performed to compare the goodness of fit of each SV using a binomial and a beta-binomial distribution. We found that the binomial distribution adequately fit most (89%) SVs. In addition, our variational formulation mitigates potential overdispersion by producing a similar effect to a beta-binomial model. The assignment probability is computed as an expectation of the binomial distribution with respect to the posterior CCF distribution; therefore, the uncertainty within the probability of success is integrated out when making assignments. Uncertainty is normally distributed in our model, while being beta-distributed in a beta-binomial model—the benefit of our choice is a fully tractable variational approximation in which all its parameters can be efficiently estimated. In the beta-binomial case, the key overdispersion parameter is difficult to estimate at typical depths obtained in whole-genome sequencing (~50x). This difficulty is evident in the high variance and lack of clear convergence observed in PyClone's MCMC traces of its overdispersion parameter (Supplementary Fig. 11).

**Posterior inference**. We estimate the unknown $\phi_k$ and $m_{i,j}$ in Eq. (5) by variational inference (VI). Specifically, the algorithm obtains a posterior distribution over $\phi_k$ and a point estimate of $m_{i,j}$. For $\phi_k$, we specify a Gaussian distribution as its prior. The choice is motivated by its convenience in deriving a fully trackable VI method. As a result, we obtain a maximum pseudo marginal likelihood estimator for $m_{i,j}$. The model employs a finite mixture model, hence, we introduce additional parameters such as the mixing coefficient $\pi_k$ and the cluster assignment variable $z_{j,k}$, which have the standard Dirichlet and Categorical prior respectively. We use this formulation for both clonal and subclonal copy-number settings. In regions of clonal copy number, the mapping is exact. In the presence of subclonal copy number, the mapping is an approximation, in which $n_{tot_t}$ is replaced by the weighted average total tumour copy number. Here we provide a detailed description of the inference.

The variational inference method maximises the evidence lower bound (ELBO) of the marginal likelihood of the model:

$$logp(B|M, H) = log\int p(B|Z, \phi, M, H)p(Z|\pi)p(\phi)p(\pi)dZd\phi d\pi$$
$$= log\int p(B, Z, \phi, \pi|M, H)dZd\phi d\pi \qquad (6)$$
$$\geq E_{q(Z,\phi,\pi)}[logp(B, Z, \phi, \pi|M, H)] - E_{q(Z,\phi,\pi)}[logq(Z, \phi, \pi)],$$

where $B = \{b_{i,j}\}$, $Z = \{Z_{i,k}\}$, $\phi = \{\phi_k\}$, $\pi = \{\pi_k\}$, $M = \{m_{i,j}\}$. We use $H$ to represent all fixed variables.

Assuming independence among the unknowns, $q(Z, \phi, \pi) = q(Z)q(\phi)q(\pi)$, the ELBO is maximised by the following solution:

$$q(Z) \propto exp\left(E_{q(\phi,\pi)}[logp(B, Z, \phi, \pi|M, H)]\right)$$
$$q(\phi) \propto exp\left(E_{q(Z,\pi)}[logp(B, Z, \phi, \pi|M, H)]\right) \qquad (7)$$
$$q(\pi) \propto exp\left(E_{q(Z,\phi)}[logp(B, Z, \phi, \pi|M, H)]\right)$$

further details about these approximations can be found in[26].

The multiplicities are estimated by the following maximisation formula:

$$\widehat{m}_{i,j} = argmax_{m_{i,j}\in\Xi_{i,j}}\sum_{k=1}^{K} E_{q(z_{i,k},\phi_k)}\left[logp\left(b_{i,j}|d_{i,j}, f_{i,j,k}, z_{i,k} = 1\right)\right] \qquad (8)$$

The difference between clonal and subclonal copy number is reflected in the set of possible multiplicities:

$$\Xi_{i,j} = \begin{cases} \{1, ... n_{maj_t,i,j}\}, & \text{if tumour copy number at SV } i, j \text{ is clonal} \\ \{\sum_{r=1}^{2}\rho_{r,i,j}x_{i,j} : x_{i,j} \in 0, ... n_{maj_t,r,i,j}\}, & \text{if tumour copy number at SV } i, j \text{ is subclonal.} \end{cases} \qquad (9)$$

with $\Xi_{i,j} = \{1, ... n_{maj_t,i,j}\}$ if the tumour copy-number segment at $SV_{i,j}$ is clonal, and $\Xi_{i,j} = \{\sum_{r=1}^{2}\rho_{r,i,j}m_{i,j} : m_{i,j} \in 0, ... n_{maj_t,r,i,j}\}$ if the tumour copy-number segment at $SV_{i,j}$ is subclonal. Where $n_{maj_t,i,j}$ is the tumour major copy number at $SV_{i,j}$. $\rho_{r,i,j}$ and $n_{maj_t,r,i,j}$ are the fraction and major copy number of the tumour subclonal $r$ at $SV_{i,j}$.

The variational inference algorithm is run over a range of possible cluster numbers (by default 1..6) and multiple repeats (by default 5). The solution with the best ELBO is selected.

**Calculating variant CCFs**. Given the estimated multiplicity, $m_{i,j}$, we infer CCF at individual variant level. In our Binomial model, the probability of observing a variant supporting read is specified as $f_{i,j,k} = w_{i,j}\phi_k + \epsilon$. We replace the cluster-level CCF parameter $\phi_k$ with variant-level CCF parameter $\phi_{i,j}$. As a result, we have $\phi_{i,j} = \frac{f_{i,j}-\epsilon}{w_{i,j}}$. The removal of subscript $k$ in $f_{i,j}$ reflects the change in CCF levels. Under the Binomial distribution assumption for variant supporting read counts, $f_{i,j} = E[VAF_{i,j}]$, $VAF_{i,j}$ is an unbiased estimator of $f_{i,j}$. Therefore, $\phi_{i,j}$ can be estimated as $\frac{VAF_{i,j}-\epsilon}{w_{i,j}}$. Note that, the linear relationship doesn't support a natural bound on CCF. We cap the maximum of CCF at 2. Finally, we have

$$CCF_{i,j} = min\left(2, \frac{VAF_{i,j}-\epsilon}{w_{i,j}}\right) \qquad (10)$$

For SVs, the mean of these two CCFs is used as the representative CCF per SV.

**Post-assignment of variants to clusters**. In some cases, the number of filtered SVs in a sample is too small $\lesssim 10$ to perform reliable clustering. To estimate the CCF of these SVs, we leverage clustering results from SNV data. To demonstrate this approach, we use the subscript $_{post}$ to denote variables of the post-assigned SVs, e.g. $B_{post}$. The strategy is to use $q(\phi_{SNV})$ (the subscript $_{SNV}$ emphasises that the distribution is constructed from SNVs) from Ccube results as a reference model, then assign SVs to clusters in $q(\phi_{SNV})$. Algorithmically, given that Ccube and SVclone both assume $q(\phi)$ to be Gaussian, one can use the SNV-based $q(\phi_{SNV})$ to compute $q(Z_{post})$ and $q(\pi_{post})$ for the SVs.

Here, we set out to update $q(\pi_{post})$ in addition to the assignment variable $q$ ($Z_{post}$). The reason for this is to avoid the post-assignment mimicking the mixing proportions in SNV results. The dimensions of $q(Z_{post})$ and $q(\pi_{post})$ are set to be compatible with the number of clusters in $q(\phi_{SNV})$. More precisely,

$$q(Z_{post}) \propto exp\left(E_{q(\phi_{SNV},\pi_{post})}\left[logp\left(B_{post}, Z_{post}, \phi_{SNV}, \pi_{post}|M, H_{post}\right)\right]\right)$$
$$q(\pi_{post}) \propto exp\left(E_{q(Z_{post},\phi_{SNV})}\left[logp\left(B_{post}, Z_{post}, \phi_{SNV}, \pi_{post}|M_{post}, H_{post}\right)\right]\right) \qquad (11)$$

The multiplicities of the post-assigned, $\widehat{m}_{i,j,post}$, are estimated as:

$$argmax_{m_{i,j,post}\in\Xi_{i,j}}\sum_{k=1}^{K} E_{q(z_{i,k,post},\phi_{k,SNV})}\left[logp\left(b_{i,j,post}|d_{i,j,post}, f_{i,j,k}, z_{i,k,post} = 1\right)\right] \qquad (12)$$

where the set of possible states $\Xi_{i,j}$ is of the same form with the settings in the main clustering algorithm, Eq. (9). Similar to above, and SVs that were initially filtered can be post-assigned to the SNV or SV clusters.

**Quality control**. We use the same quality-checking steps for both clustering and post-assign. They are made of three steps: (1) remove empty clusters, (2) remove small clusters with less than 1% of the data assigned, (3) merge clusters with cluster means less than 10% apart. In each step, the model parameters are refined with the same variational inference procedures.

**SV simulation**. SVs were simulated by first rearranging the reference genome to create an artificial genome containing SVs, and then simulating reads with Sim-Seq[37] from this rearranged reference. The reads were then mapped back to the original, unmodified reference genome using bowtie2 with the local alignment flag[38]. SV size was randomly chosen among the size categories 300–1 kb, 2–10 kb and 20–100 kb with equal probability for each category.

We simulated a single, heterozygous chromosome 12 (being roughly representative of genome-wide GC-content) with SVs of a single type at every 100 kb interval. Samples containing only deletions, translocations, inversions and duplications were generated at the tumour purity levels of 100, 80, 60, 40 and 20%. We generated 100 bp paired-end reads with an average fragment size of 300 bp and an insert-size standard deviation of 20 bp. The SV events were assumed to always occur in the heterozygous fashion, hence the 'true' VAF was always considered to be half of the simulated purity value. To achieve the effect of differing purities, simulated normal reads were mixed with tumour samples with coverage equivalent to $\frac{\lambda \cdot t}{2}$ and normal read coverage of $\frac{\lambda \cdot (1-t)}{2}$ where $\lambda$ represents the expected total read count at a locus. We ran simulations at 50x coverage, typical for WGS data by simulating $\frac{50L}{300}$ total reads per simulation where $L$ is the chromosome length (post rearrangement) and 300 is the fragment length. The number of reads generated for

genomes with variants that changed the total size of the genome (deletions and duplications) was adjusted by the new genome length.

The list of SV breakpoints for each simulation run was collated, directions were inferred and each breakpoint was classified using SVclone's annotate step. Any breakpoints where the direction could not be inferred correctly, or their classification was incorrect were discarded. The resulting set was run through SVclone's count step with default parameters. SVs were then filtered through the filter step, and adjusted VAF field was used to compare variant frequencies at corresponding purity levels.

**Prostate sample mixing.** The metastatic samples bM (A) and gM (B) from Patient 001[23] were chosen due to their similar coverage (51.5x and 58.9x) and purity (49 and 46%). Previous analysis by Hong et al.[23] showed that these metastases shared a common ancestral clone, had no evidence of subclonality, and contained a number of private SVs and SNVs. Mixing two clonal metastases from the same patient has many advantages over spike-in approaches including: realistic sequencing noise, realistic subclonal mixing of SVs, SCNAs and SNVs, and a natural branching clonal architecture with both clonal and subclonal mutations present. We generated a total of nine samples with subclonal mixes of reads sampled at percentages 10–90, 20–80, 30–70, 40–60, 50–50, 60–40, 70–30, 80–20, and 90–10 for metastasis A and B, respectively. Three clusters are expected to be revealed upon mixing: shared variants present at 100% CCF, one cluster at bM's mixture frequency and one cluster at gM's mixture frequency. We also generated mixtures of four and five clusters each. The four cluster mixture was constructed by subsampling bM's odd and even chromosomes separately at 20 and 60% respectively, and then mixing this with a 40% subsampled mixture from gM's odd chromosomes only (effectively creating a mixture where odd and even chromosomes comprise 60% of either or both samples, with CCFs of 20%, 40 and 60%). Similarly, the five-cluster mixture was constructed by subsampling bM's odd and even chromosomes separately at 80 and 60% respectively, and gM's odd and even chromosomes at 20 and 40% respectively (effectively creating a mixture where odd and even chromosomes comprise 100% of both samples, with CCFs of 20, 40 and 60 and 80%).

A merged variant list was created for SVs and SNVs, containing both the individual sample's high-confidence calls. SV breakpoints were then run through SVclone's complete pipeline, and SNVs were counted at each variant locus using Samtool's mpileup[39] and pileup2base (https://github.com/riverlee/pileup2base) for each mixture. Battenberg was run on each mixture to obtain SCNA data and purity estimates (which were used as the purity values for both SVclone and PyClone). A truth set was created for benchmarking purposes, constructed for SV, SNV and SCNA by determining whether the variant was unique to one sample, or shared in both.

In silico mixtures were created using the subsample and merge functions from SAMtools v1.2. copy numbers were obtained from Battenberg on each merged sample with default parameters. To construct the breakpoint list for input into SVclone's annotate step, Socrates was run on the individual bM and gM samples, then run through SVclone's annotate and count steps (using Socrates' directions, filtered on simple and satellite repeats using the repeat-masker track (repeatmasker.org) and a minimum average MAPQ of 20). The resulting bM and gM SVs were then merged and filtered against the germline. copy numbers were matched using corresponding Battenberg subclonal copy-number output. The merged SV list was used as the set of SV calls for the annotate step for each mix.

The reference and variant alleles were counted at each of the 9810 SNVs across the different mixture proportion BAM files (Mutect variant calls from Hong et al. were used with alleles recounted using Samtool's mpileup and pileup2base (https://github.com/riverlee/pileup2base) using a minimum quality and MAPQ cutoffs of 20 to count a base. Battenberg was run on each mixture and was used to provide copy-number information for each variant locus, as well as the purity estimate for both SVclone and PyClone. For SNV clustering, we filtered out any variants in regions of subclonal copy number (PyClone does not support subclonal copy-number handling). To improve performance, cluster labels from the PyClone traces were obtained using the mpear function from the mcclust R package https://cran.r-project.org/web/packages/mcclust. Variant and cluster CCFs were calculated from the mean MCMC trace values. We subsampled 5000 variants from the resulting SNV output per mixture and ran these variants through the PyClone algorithm for 2500 iterations with a burn-in of 1000.

Set ownership of SVs (whether the SV is present in bM only, gM only or shared between the two), was determined by running the union list of variants through SVclone against each 001 sample. If there were any supporting reads for both samples, the SV was considered shared, otherwise it was considered unique to the sample that it exclusively appeared in. For the SNVs, a variant was considered exclusive to a sample if it was called in one individual sample's consensus SNV list only, and shared if it appeared in both lists. To test set ownership of SCNAs, Battenberg calls run on bM and gM were analysed. Any SCNAs present in both bM and gM (where start and end coordinates had to be at least within 5 kb of each other) that contained the same allelic copy numbers were considered shared SCNAs. Any SCNAs with partial matches (only one end matched, or copy numbers differed) were discarded. All other SCNAs from considered unique to their respective samples. The mixture SCNAs were then compared with this list (where both ends must match within a 5 kb boundary). The SCNA fraction that matched the given sample's allelic copy number was used as the SCNA's CCF estimate. To

calculate mean cluster CCF error, we compared: (i) highest ground truth CCF to highest derived cluster CCF, (ii) lowest ground truth CCF to lowest derived cluster CCF, (iii) second-highest ground truth CCF etc. alternating between highest and lowest in ranked order until either there are no more derived clusters or no more truth clusters.

**Determining optimal variant multiplicities and CCFs.** To calculate the optimal CCF per variant across all mixes, we took the true mixture state of the variant, and inferred the best multiplicity. For example, in the 70–30 mixture, if an SV was present only in bM and not in gM, the true cluster CCF of 0.7 was used as the $\phi_k$ when calculating $f_{i,j,k}$ (the binomial probability of sampling a variant read for the given locus). The multiplicity was then inferred using Eq. (6). An adjusted variant CCF could then be determined using the same method outlined in the section titled calculating variant CCFs.

**Testing the distribution fit of SV allele frequencies.** To test whether a binomial model was appropriate for modelling SV VAFs, SVclone's annotate to filter steps were run on the bM and gM metastases from patient 001, retaining only variants with copy-number neutral states (1/1 for major/minor allelic copy numbers). For each variant, a binomial likelihood for the observed number of supporting reads was calculated using $p_j = t/2$ (the theoretical heterozygous variant frequency) and $d_j$ equivalent to the observed variant's adjusted read depth. A beta-binomial likelihood was also calculated where the $\beta$ value was empirically estimated from the data as:

$$\beta = \frac{(\mu - n)(\mu^2 - \mu n + \sigma^2)}{\mu^2 - \mu n + n\sigma^2},$$ (13)

where $\mu = \mu_d(t/2)$ (mean adjusted depth multiplied by the theoretical heterozygous variant frequency) and $\sigma$ is the standard deviation of the observed supporting reads. $\alpha$ was then estimated as:

$$\alpha = -\frac{\mu\beta}{\mu - d_j}$$ (14)

A likelihood-ratio test was then applied with one degree of freedom to each variant with the binomial distribution as the null model, and the beta-binomial as the alternative. Of 55 SVs tested in the bM sample, 6 SVs rejected the null model (10.9%); 2 of 44 SVs (4.5%) rejected the null model in the gM sample. Therefore, >89% SVs of moderate purity and coverage appeared to be consistent with the binomial model, indicating that the binomial model is a reasonable choice of distribution for SV data given the moderate coverage and purity of the analysed datasets.

**Copy-number perturbation experiments.** We selected the 70–30 in silico mixture due to its low mean variant CCF error for the SCNA perturbation experiments. In order to perform experiments representative of the background copy-number heterogeneity prevalence, we quantified the per-sample fraction of SVs that had different background copy-number states across PCAWG. Supplementary Table 1 includes the medians of this measure cohort-wide and by histology group. We observed a range of background copy-number heterogeneity across the cohort, with a minimum (median) of 34% in non-Hodgkin lymphoma and a maximum (median) of 74% in colorectal adenoma. The median across all cancer types was 0.53. Given this result, we investigated the same measure in the three-cluster in silico mixtures and identified a lower background SCNA heterogeneity rate (potentially due to SCNA averaging as background SCNA heterogeneity was higher in the pure 001 bM and 001 gM samples) (see Supplementary Table 6). We therefore randomly removed SVs with homogeneous background SCNAs until the rate of heterogeneity was 50%, resulting in 45 SVs with homogeneous and heterogeneous background SCNA states, and used these data for downstream experiments (see the SVclone_Rmarkdown notebook under code availability to replicate this analysis).

SCNAs were perturbed as follows: (i) CN − 1: major alleles were subtracted by one in the fraction A subclone in Battenberg. If the copy number was 1, subtracting one from the minor allele was attempted. If the copy-number state was 1-0, no modification was performed (only two SVs were unable to be changed); (ii) CN + 1: major alleles were incremented by one for the fraction A clone; (iii) Frac ± 0.3: 0.3 was added to the SCNA fraction for subclonal SVs, unless the modified fraction were to exceed 0.9, in which case 0.3 was subtracted. The new SCNA fraction of clone B was calculated as one minus the new fraction of clone A.

The above experiments perturbed the $l$ side of each SV for the one-side experiments, and both the $l$ and $u$ sides for both-side experiments. Performance metrics were calculated as usual with the two single-end metrics averaged out between the two runs (Supplementary Fig. 3).

**Analysis of ICGC/TCGA pan-cancer samples.** We utilised the pan-cancer analysis of whole genomes (PCAWG) October 12th 2016 consensus SNV call set, the v1.6 consensus SVs and the consensus subclonal copy numbers (19th of January 2017). For a detailed explanation on how these were generated, see[25]. Annotate and count were run using each sample's associated mini-bam. Consensus purity and ploidy estimates (January 9th 2017) were used. Sample SVs and SNVs were run

separately through SVclone's SV and SNV clustering model with default parameters.

We considered only white-listed PCAWG samples that had sufficient power to detect subclonality (number of reads per chromosome copy or NRPCC > 10; $n = 1705$, see Supplementary Note 1 for a list of samples). As a QC measure, we tested the association of SV number with sample purity (Supplementary Fig. 12), and found the variables to be uncorrelated ($R^2 = 0.001$). We also tested the rate at which SVclone called single non-clonal clusters in PCAWG samples. Using a cutoff of <0.7 cluster CCF, and considering only samples with >10 variants (of the clustered variant type) the SV clustering reported one sample with single non-clonal clusters across 1220 PCAWG samples with (0.0008%), and the SNV clustering reported four samples across 1362 (0.0029%) samples. These results indicated that rates of under-clustering were low, and were similar across the SV and SNV clustering models.

We tested each PCAWG sample for the enrichment of balanced rearrangements (inversions and inter-chromosomal translocations) below the CCF cutoff (0.7) using a hypergeometric test, with the alternative hypothesis of P(X ≥ x), where $x = \sum_{j=1}^{J} = 1[\kappa_j = \kappa_{bal}]$ and $\kappa_j$ refers to a given SV's classification. Survival analysis was undertaken using the *survival* CRAN package (cran.r-project.org/ package = survival). Hazard ratios were calculated using the Cox proportional hazards regression model, stratified by tier 4 tumour histology, age and the number of SVs in 1–100, 101–200 etc. bins. We used a hypergeometric test to determine whether any ICGC/TCGA contributors were overrepresented for SCNR samples within each histology type and found no evidence of any significant over-representation (using an FDR < 0.05 significance threshold). To determine whether SCNR samples were overrepresented for fold-back inversions (FBI) in the ovarian samples, we used a one-sided $t$ test to compare SCNR samples to the high SV heterogeneity and other subsets, and found no significance ($p = 0.8056$ and $p = 0.4671$, respectively). Supplementary Fig. 6 shows a boxplot of amplified FBI fraction across the three subsets. For the clustering of breakpoints criteria, SVs were tested on a per-chromosome basis (inter-chromosomal SVs were removed). The ability to walk each derivative chromosome was tested using criteria for chromothripsis tests A and F[29]. Chromosomes were only tested if they contained at least four clonal and four subclonal rearrangements per chromosome.

A list of consensus coding driver genes was obtained from the curated PCAWG coding driver genes (29th of September 2016). Patient-centric coding point mutations were obtained from Sabarinathan et al. Table S2[40]. CCFs were matched using SVclone's clustering results, and variants with a CCF < 0.7 were considered subclonal. Enrichment of driver genes was computed using a hypergeometric test for the SCNR and high SV heterogeneity cohorts using the driver mutations from all 1705 samples as the complete sample set. Copy numbers were obtained from the PCAWG annotated consensus clonal copy numbers (19th of January 2017). Loss of heterozygosity was defined as any region where the minor allele was zero (X and Y chromosomes in males were only considered in cases of complete loss). Copy-number gains or amplifications were not considered. SV driver hits were defined as any SV that affected at least one exon of a driver gene, and was not completely outside the gene (i.e. at least one SV break-end must fall within the gene body. Any regions with deletions (called as structural variants) that were also affected by copy-number loss (called from copy number) were considered as one variant only to avoid redundancy.

**Reporting summary**. Further information on research design is available in the Nature Research Reporting Summary linked to this article.

## Data availability

In silico sample mixtures were generated from patient data derived from patient 001 from the Hong et al. study[23]. The data are available in the EGA Sequence Read Archive under accession EGAS00001000942.

Somatic and germline variant calls, mutational signatures, subclonal reconstructions, transcript abundance, splice calls and other core data generated by the ICGC/TCGA Pan-cancer Analysis of Whole Genomes Consortium is described in ref [20] and available for download at https://dcc.icgc.org/releases/PCAWG. Additional information on accessing the data, including raw read files, can be found at https://docs.icgc.org/pcawg/data/. In accordance with the data access policies of the ICGC and TCGA projects, most molecular, clinical and specimen data are in an open tier which does not require access approval. To access potentially identification information, such as germline alleles and underlying sequencing data, researchers will need to apply to the TCGA Data Access Committee (DAC) via dbGaP (https://dbgap.ncbi.nlm.nih.gov/aa/wga.cgi?page=login) for access to the TCGA portion of the dataset, and to the ICGC Data Access Compliance Office (DACO; http://icgc.org/daco) for the ICGC portion. In addition, to access somatic single-nucleotide variants derived from TCGA donors, researchers will also need to obtain dbGaP authorisation. Derived datasets described specifically in this manuscript can be found at these locations:

https://www.synapse.org/#!Synapse:syn7596712 (consensus SVs)
https://www.synapse.org/#!Synapse:syn7357330 (consensus SNVs and INDELs)
https://www.synapse.org/#!Synapse:syn8042880 (consensus copy-numbers)

All the other data supporting the findings of this study are available within the article and its supplementary information files and from the corresponding author upon reasonable request. A reporting summary for this article is available as a Supplementary Information file.

## Code availability

The SVclone software, user documentation, and example data can be downloaded from https://github.com/mcmero/SVclone. Ccube clustering code can be found under https://github.com/keyuan/ccube. Code for generating all figures in the manuscript and the in silico mixture samples can be found under https://github.com/mcmero/SVclone_Rmarkdown. Code for simulating SVs can be found under https://github.com/mcmero/sv_simu_pipe.

The core computational pipelines used by the PCAWG Consortium for alignment, quality control and variant calling are available to the public at https://dockstore.org/search?search=pcawg under the GNU General Public License v3.0, which allows for reuse and distribution.

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

## Acknowledgements
We would like to thank Kangbo Mo for his assistance in developing the SV classification scheme during his Master studies, and Christoffer Flensburg for his helpful advice for the four- and five-cluster mixtures. F.M., G.M. and K.Y. would like to acknowledge the support of the University of Cambridge, Cancer Research UK and Hutchison Whampoa Limited. G.M., K.Y. and F.M. were funded by CRUK grants C14303/A17197 and A19274. G.M. was funded by CRUK grant A15973. This work was supported, in part, by NHMRC grants 1047581 and 1104010 to C.M.H. and 1024081 to N.M.C as well as a VCA early career seed grant 14010 to NMC. NMC was supported by a David Bickart Clinician Researcher Fellowship from the Faculty of Medicine, Dentistry and Health Sciences, University of Melbourne, and more recently by a Movember – Distinguished Gentleman's Ride Clinician Scientist Award through the Prostate Cancer Foundation of Australia's Research Program. M.C. would like to acknowledge the support of the Cybec Foundation and the Endeavour Research Fellowship. We acknowledge the contributions of the many clinical networks across ICGC and TCGA who provided samples and data to the PCAWG Consortium, and the contributions of the Technical Working Group and the Germline Working Group of the PCAWG Consortium for collation, realignment and harmonised variant calling of the cancer genomes used in this study. We thank the patients and their families for their participation in the individual ICGC and TCGA projects.

## Author contributions
M.C.: methodology, analysis, software, visualisation, manuscript writing, editing and review; K.Y.: methodology, software, manuscript writing, editing and review; C.S.O.: methodology; J.S.: methodology; N.M.C.: supervision, manuscript editing and review; T.P.: supervision, methodology; C.M.H.: supervision, manuscript editing and review; F.M.: methodology, supervision, manuscript editing and review; G.M.: supervision, conceptualisation, methodology, analysis, manuscript writing, editing and review. The P.C.A.W.G. Evolution and Heterogeneity Working Group (led by P.S., PvL and D.C.W.): analysis.

## Competing interests
R.B. owns equity in Ampressa Therapeutics. G.G. receives research funds from IBM and Pharmacyclics and is an inventor on patent applications related to MuTect, ABSOLUTE, MutSig, MSMuTect and POLYSOLVER. I.L. is a consultant for PACT Pharma. B.J.R. is a consultant at and has ownership interest (including stock, patents, etc.) in Medley Genomics. All the other authors have no competing interests.

## Additional information

## PCAWG Evolution and Heterogeneity Working Group

David J. Adams[11], Pavana Anur[12], Rameen Beroukhim[13,14,15], Paul C. Boutros[16,17,18,19], David D.L. Bowtell[20,21], Peter J. Campbell[11,22], Shaolong Cao[23], Elizabeth L. Christie[20], Yupeng Cun[24], Kevin J. Dawson[11], Jonas Demeulemeester[25,26], Stefan C. Dentro[11,26,27], Amit G. Deshwar[28], Nilgun Donmez[29,30], Ruben M. Drews[31], Roland Eils[32,33,34,35], Yu Fan[23], Matthew W. Fittall[26], Dale W. Garsed[20,36], Moritz Gerstung[37,38], Gad Getz[13,15,39,40], Santiago Gonzalez[37,38], Gavin Ha[13], Kerstin Haase[26], Marcin Imielinski[41,42], Lara Jerman[38,43], Yuan Ji[44,45], Clemency Jolly[26], Kortine Kleinheinz[32,34], Juhee Lee[46], Henry Lee-Six[11], Ignaty Leshchiner[13], Dimitri Livitz[13], Salem Malikic[29,30], Iñigo Martincorena[11], Thomas J. Mitchell[11,47,48], Quaid D. Morris[49,50], Ville Mustonen[51,52,53], Layla Oesper[54], Martin Peifer[24],

Myron Peto[55], Benjamin J. Raphael[56], Daniel Rosebrock[13], Yulia Rubanova[50,57], S. Cenk Sahinalp[29,30,58], Adriana Salcedo[16], Matthias Schlesner[32,59], Steven E. Schumacher[13,60], Subhajit Sengupta[61], Ruian Shi[49], Seung Jun Shin[62], Paul T. Spellman[63], Oliver Spiro[13], Lincoln D. Stein[16,64], Maxime Tarabichi[11,26], Peter Van Loo[25,26], Shankar Vembu[49,65], Ignacio Vázquez-García[11,66,67,68], Wenyi Wang[23], David C. Wedge[11,30,69], David A. Wheeler[70,71], Jeffrey A. Wintersinger[50,72,73], Tsun-Po Yang[27], Xiaotong Yao[44,74], Kaixian Yu[75] & Hongtu Zhu[76,77]

[11]Wellcome Sanger Institute, Wellcome Genome Campus, Hinxton, Cambridge CB10 1SA, UK. [12]Molecular and Medical Genetics, Oregon Health and Science University, Portland, OR 97201, USA. [13]Broad Institute of MIT and Harvard, Cambridge, MA 02142, USA. [14]Department of Medical Oncology, Dana-Farber Cancer Institute, Boston, MA 02115, USA. [15]Harvard Medical School, Boston, MA 02115, USA. [16]Computational Biology Program, Ontario Institute for Cancer Research, Toronto, ON M5G 0A3, Canada. [17]Department of Medical Biophysics, University of Toronto, Toronto, ON M5S 1A8, Canada. [18]Department of Pharmacology, University of Toronto, Toronto, ON M5S 1A8, Canada. [19]University of California Los Angeles, Los Angeles, CA 90095, USA. [20]Peter MacCallum Cancer Centre, Melbourne, VIC 3000, Australia. [21]Sir Peter MacCallum Department of Oncology, University of Melbourne, Melbourne, VIC 3052, Australia. [22]Department of Haematology, University of Cambridge, Cambridge CB2 2XY, UK. [23]Department of Bioinformatics and Computational Biology, The University of Texas MD Anderson Cancer Center, Houston, TX 77030, USA. [24]University of Cologne, 50931 Cologne, Germany. [25]University of Leuven, B-3000 Leuven, Belgium. [26]The Francis Crick Institute, London NW1 1AT, UK. [27]Big Data Institute, Li Ka Shing Centre, University of Oxford, Oxford OX3 7LF, UK. [28]The Edward S. Rogers Sr. Department of Electrical and Computer Engineering, University of Toronto, Toronto, ON M5S 3G4, Canada. [29]Simon Fraser University, Burnaby, BC V5A 1S6, Canada. [30]Vancouver Prostate Centre, Vancouver, BC V6H 3Z6, Canada. [31]Cancer Research UK Cambridge Institute, University of Cambridge, Cambridge CB2 0RE, UK. [32]Division of Theoretical Bioinformatics, German Cancer Research Center (DKFZ), 69120 Heidelberg, Germany. [33]Heidelberg University, 69120 Heidelberg, Germany. [34]Institute of Pharmacy and Molecular Biotechnology and BioQuant, Heidelberg University, 69120 Heidelberg, Germany. [35]New BIH Digital Health Center, Berlin Institute of Health (BIH) and Charité - Universitätsmedizin Berlin, 10117 Berlin, Germany. [36]Sir Peter MacCallum Department of Oncology, The University of Melbourne, Melbourne, VIC 3052, Australia. [37]European Molecular Biology Laboratory, European Bioinformatics Institute (EMBL-EBI), Wellcome Genome Campus, Hinxton, Cambridge CB10 1SD, UK. [38]Genome Biology Unit, European Molecular Biology Laboratory (EMBL), 69117 Heidelberg, Germany. [39]Center for Cancer Research, Massachusetts General Hospital, Boston, MA 02129, USA. [40]Department of Pathology, Massachusetts General Hospital, Boston, MA 02115, USA. [41]New York Genome Center, New York, NY 10013, USA. [42]Weill Cornell Medicine, New York, NY 10065, USA. [43]University of Ljubljana, 1000 Ljubljana, Slovenia. [44]Research Institute, NorthShore University HealthSystem, Evanston, IL 60201, USA. [45]Department of Public Health Sciences, The University of Chicago, Chicago, IL 60637, USA. [46]Department of Statistics, University of California Santa Cruz, Santa Cruz, CA 95064, USA. [47]University of Cambridge, Cambridge CB2 1TN, UK. [48]Cambridge University Hospitals NHS Foundation Trust, Cambridge CB2 0QQ, UK. [49]University of Toronto, Toronto, ON M5G 2M9, Canada. [50]Vector Institute, Toronto, ON M5G 0A3, Canada. [51]Department of Computer Science, University of Helsinki, 00014 Helsinki, Finland. [52]Institute of Biotechnology, University of Helsinki, 00014 Helsinki, Finland. [53]Organismal and Evolutionary Biology Research Programme, University of Helsinki, 00014 Helsinki, Finland. [54]Department of Computer Science, Carleton College, Northfield, MN 55057, USA. [55]Molecular and Medical Genetics, Oregon Health & Science University, Portland, OR 97239, USA. [56]Department of Computer Science, Princeton University, Princeton, NJ 08540, USA. [57]Department of Computer Science, University of Toronto, Toronto, ON M5S 1A8, Canada. [58]Indiana University, Bloomington, IN 47405, USA. [59]Bioinformatics and Omics Data Analytics, German Cancer Research Center (DKFZ), 69120 Heidelberg, Germany. [60]Department of Cancer Biology, Dana-Farber Cancer Institute, Boston, MA 02215, USA. [61]Center for Psychiatric Genetics, NorthShore University HealthSystem, Evanston, IL 60201, USA. [62]Korea University, Seoul 02481, South Korea. [63]Molecular and Medical Genetics, Knight Cancer Institute, Oregon Health & Science University, Portland, OR 97219, USA. [64]Department of Molecular Genetics, University of Toronto, Toronto, ON M5S 1A8, Canada. [65]Argmix Consulting, North Vancouver, BC V7M 2J5, Canada. [66]Department of Applied Mathematics and Theoretical Physics, Centre for Mathematical Sciences, University of Cambridge, Cambridge CB3 0WA, UK. [67]Department of Epidemiology and Biostatistics, Memorial Sloan Kettering Cancer Center, New York, NY 10065, USA. [68]Department of Statistics, Columbia University, New York, NY 10027, USA. [69]Oxford NIHR Biomedical Research Centre, University of Oxford, Oxford OX4 2PG, UK. [70]Department of Molecular and Human Genetics, Baylor College of Medicine, Houston, TX 77030, USA. [71]Human Genome Sequencing Center, Baylor College of Medicine, Houston, TX 77030, USA. [72]The Donnelly Centre, University of Toronto, Toronto, ON M5S 3E1, Canada. [73]Department of Computer Science, University of Toronto, Toronto, ON M5S 2E4, Canada. [74]Tri-institutional PhD Program of Computational Biology and Medicine, Weill Cornell Medicine, New York, NY 10065, USA. [75]Department of Biostatistics, The University of Texas MD Anderson Cancer Center, Houston, TX 77030, USA. [76]Department of Biostatistics, University of North Carolina at Chapel Hill, Chapel Hill, NC 27599, USA. [77]The University of Texas MD Anderson Cancer Center, Houston, TX 77030, USA

