## [Peer Review File · Nature Communications]

Reviewers' comments:

Reviewer #1 (Remarks to the Author):

The paper introduces a novel computational tool, SVclone, for inferring cancer-cell fraction of structural variants from DNA sequencing data. Despite the fact that problem being considered is very important, most of the existing studies and tools for reconstructing tumour subclonal composition are based on the use of SNVs and/or CNAs, with many of them inferring trees of tumour evolution. In this work, through the in silico modeling of heterogeneous tumours consisting of 2 subclonal populations (derived as a linear combination from two homogenous tumour samples of real sequencing data of the same patient), the authors demonstrate the potential use of SVclone on real sequencing data, although with moderately accurate results (for example, the average mean absolute error in estimating CCF was over 10%) which can, in part, be contributed to the sequencing coverage. SVclone is also applied on data from PCAWG project where the goal was binary classification of SVs and SNVs as clonal or subclonal. Some interesting, although not verified (which is probably impossible considering the available data), findings were obtained. Discussion of the coverage of data in PCAWG cohort is missing, as well as the overall impact of coverage on the performance of the tool. I am also sceptical about parameter setting in BIC and their effect of tool performance on data of different sequencing characteristics. The last, together with the tool's performance in the cases where tumour is composed of >3 clusters and considering different read depths, should be more thoroughly analyzed via simulated data.

Before being able to proceed to more detailed judgement of methodological contribution of the tool and testing its software, in addition to the above comments, I would also like that the following are addressed (comments are not classified as minor or major):

1. Sentence "The mutations belonging to each clone in a tumour can be interrogated using bulk whole-genome sequencing, with mutation detection subject to sequencing depth, tumour cellularity, clonality and mutation copy-number" should be made less explicit as there are other factors affecting mutation calling (for example, sequencing quality).
2. Description in Figure 1B is unclear. What do different colors (red, green, blue, grey) represent? The meaning of the last column in this figure ("adjust normal reads" etc.) isn't it obvious.
3. While presenting the error in inferring clonal frequencies in results section, it would be more convenient for the reader to interpret results if mean absolute error (MAE) is presented rather than mean squared error difference.
4. In the sentence "Both SVclone and PyClone had little difficulty in identifying minor subclonal clusters in the mixtures due to the lack of overlapping CCFs from clonal or major subclonal clusters." the second part of the sentence ("due to the lack ...") is not very clear. Could you please clarify.
5. How does the performance of SVclone compare to Pyclone in terms of number of inferred clusters and MAE if it is run in coclustering mode (i.e. using both SVs and SNVs) on in-silico mixtures of real tumour data presented in Figure 1D and Figure 1E?
6. What tumour purity values were supplied to SVclone and (if any) Pyclone while analyzing abovementioned in-silico mixtures?
7. In the model selection part in the supplementary, the authors state "In selecting the best run, we also do not consider any runs where there is a single non-clonal cluster". What is the fraction of times (on, say, 100 different runs of the tool) where SVclone reports only 1 subclonal cluster in the

analysis of in-silico mixture? Was this one of the reasons that tool correctly identified number of clusters? Note that this is also correlated to the above critics of testing the tool and its performance in the presence of more than 2 subclonal clusters since at this point it is unclear whether SVclone is under-clustering.

8. In "with >10 SVs and SNVs" please make it very clear whether "SNV + SV > 10" or "SNV>10 and SV>10"

9. In Figures 2E and 2F why there is no clear "border" between the right end of subclonal CCFs (Figure 2E) and left end of CCFs of clonal SV events (Figure 2F)? Is this a consequence of applied methodology or something else?

10. Recently, a tool for inferring trees of tumour evolution by the use of SVs was made available and accompanying method description presented in the manuscript that can be found at <https://www.biorxiv.org/content/early/2018/01/30/257014>. While I am aware that this is still not the peer-reviewed publication, considering the lack of methods for the particular problem, I would appreciate if the authors can compare against this method or justify if they are unable or unwilling to compare for whatever reason(s).

Reviewer #2 (Remarks to the Author):

This paper tackles the problem of inferring cancer cell fraction of structural variations in cancer genomes through development of a computational approach. The authors apply their approach to the PCAWG dataset - likely the largest collection of bulk whole genome cancer genomes sequenced in the world. The authors should be commended for completing an analysis of such a large dataset. The main conclusions of the paper are: the method works effectively demonstrated by simulated and real data-based validation; and through application of the method to PCAWG, identification of sub-clonal copy-neutral rearrangements as associated with inferior survival across three cancer types. The main strength of the paper is the application of SV CCF to the PCAWG dataset.

Unfortunately the remainder of the paper lacks depth and suffers from superficial treatment of the topic. Notably, the approach underlying the method itself is fundamentally not novel and includes repurposing of a previously published statistical model, albeit on SVs (which is the new part). In addition, the authors fail to adequately compare their method to previous approaches and the benchmarking they do lacks key quantitative metrics. Finally, the association of sub-clonal balanced rearrangements with survival is only superficially treated, lacks validation and lacks appropriate assessment of possible confounding co-variables. As such, the paper represents neither a convincing computational methods paper, nor a convincing biological discovery paper. My recommendation to the authors is to rework the paper considerably to either present a convincing computational methods paper with rigorous and thorough benchmarking to published methods in the field, or to undertake the appropriate scientific diligence to further substantiate the sub-clonal balanced rearrangements prognostic association. The latter should include external cohort validation, and some insight into mechanism.

Specific critique:

Proper benchmarking needs to be undertaken, comparing to other methods in the field. In particular THetA2, TITAN, Battenberg (compared to by authors in supplement), ReMixT and CloneHD are available methods that represent state of the art of copy number CCF. None of these methods tackle the specific problem of balanced rearrangement CCF, however they all quantify CCFs of other SVs. How does SVclone compare to these methods on the classes of SVs considered by all methods? If equivalency of performance can be shown, then it would be far more convincing that SVclone provides a meaningful advance to the field.

The benchmarks themselves would benefit from both calculating error on CCF as presented but also on the clustering of events. This is quite straightforward to do on both simulated and the in silico mixtures and must constitute a part of the benchmarking since clustering is a fundamental component of the method. Estimating the means of the clustering is not equivalent to ascertaining whether events are correctly co-clustered.

The statement 'One major advantage of using SVclone is that it can also be used in co-clustering mode, where CCF estimates can be given for SVs and SNVs simultaneously'. This is not substantiated with any data or analysis. The authors should show quantitatively the benefit of co-clustering in order to include such a statement, perhaps through exposing the increase in accuracy as a function of incremental inclusion of SNVs.

The authors present a finding related to breast, liver and ovarian cancer. All three tumor types have had recently described genome-based subtypes. The authors must account for these covariates in their analysis. It is well-established that the poor-outcome basal subtype of breast cancer harbours increased genomic instability. The Nik-Zainal paper describing 560 breast cancers (doi:10.1038/nature17676) has accompanying hormone receptor status so at the very least, the authors should be able to determine if their finding is independent of ER negativity, basal gene expression subtype and/or global metrics of genome instability. For ovarian cancers the Wang et al study (doi:10.1038/ng.3849) shows an enrichment of fold-back inversions in a poor prognostic group. They validate on the ICGC dataset and identify cases with fold-back inversion enrichment. Is the finding of sub-clonal balanced rearrangements independent of this?

What is the relationship of #s of SVs to the tumor content of the sample? How does this impact the ascertainment of subclonal CCF? How does tumor content influence the outcome associations as a co-variate?

What is the impact of sequencing depth on sensitivity to low CCF SVs? This and the preceding question are essential analyses to be presented to the readers.

The statement 'clonality of balanced genome rearrangements reveals functionally important and clinically relevant observations.' is not corroborated with data. The authors fail to identify any functional data or experiments to support this statement.

The statement 'Importantly, considering only the clonality of SNVs and/or SCNAs would have failed to reveal this information.' is unsupported by any analysis. It is entirely possible that SCNAs at similar CCF represent clones that are the latent cause for poor outcome. SCNRs might simply be part of the genomic landscape of the clone that has a phenotype. The authors would have to demonstrate the the SCNRs are actually independent of SCNAs at similar CCF to substantiate this claim.

The authors somehow seem to miss an opportunity to associate their findings with driver mutations in these genomes as a route to indicating mechanism for SCNRs. Are there significant associations to the prevalence of driver mutations in the cases harboring SCNRs?

The method itself relies heavily on the PyClone probabilistic model (doi:10.1038/nmeth.2883), repurposed for clustering SVs. This is not properly acknowledged in the main text anywhere and represents a major oversight/omission. (I note it is referenced in the supplementary info). Also the graphical model (very similar to PyClone) is never referenced in the main, nor supplementary text. If the authors are claiming novelty, they should establish how their method differs from the PyClone approach.

The authors rely on a linear scaling of the SV supporting reads dependent on the SV type. The authors test on a prostate sample which are known to harbour primarily blunt end joined DNA breaks. Have the authors tested the robustness of their scaling method when either

microhomology or non-templated inserted sequence is present at the breakpoint? What is the effect of read length?

Do the authors observe multi-modality in the posterior distributions of CCF for individual events? This is likely going to be the case in many situations where the relative combination of ploidy, tumor content and CCF is unidentifiable? How, in turn does this impact the distribution over clusterings of the events. The authors should show co-clustering distributions computed over the 25,000 MCMC samples.

'Clusters which have no variants assigned are discarded'. How is it possible in a DP to have an empty cluster?

There is a comparison to Battenberg CCFs presented only in the supplementary information. In Supplementary Fig 7, the authors present correlations. The data appear substantially uncorrelated across all size categories. This raises a serious concern to this reviewer and suggests quite unexpected behaviour from SVClone. Which method is correct? Why? Under what circumstances should a user trust SVClone vs Battenberg (or for that matter, TITAN, THetA2, ReMixT once those comparisons have been made)? This needs appropriate treatment using established comparison methods fitting of a computational methods paper.

The authors justification of the binomial distribution is convoluted and unconvincing. First the authors remove VAF outliers before testing for goodness of fit which surely improves the goodness of fit to a distribution with constrained variance such as the Binomial. The authors use a chi squared test which is only appropriate for categorical data, confusingly with two Poisson distributions instead of a Binomial. The authors should fit the data to both a Binomial and Beta Binomial (the distribution used by PyClone) or other distribution and use a likelihood ratio test.

The modified BIC has no theoretical justification. The main motivation for using a DP is to obviate model selection.

Minor comments

Overall the paper is quite short and contains a lot of relevant material in the supplementary info that would be of interest to the reader if presented in the main text.

There are supplementary figures not referenced in the main text.

Dear Reviewers,

We would like to thank you for your effort and time. The insightful reviews of our manuscript have vastly improved our method and manuscript.

The feedback provided raised a number of important methodological issues which prompted us to rethink our clustering approach. In the revised manuscript we present an improved clustering algorithm based on variational Bayesian inference which addresses the shortcomings of our original approach. As such, many of the initial issues raised are no longer applicable and are indicated as such in our response below. One clear methodological advance is that the new model now uses both break ends of a structural variant rather than picking a single end, improving novelty and performance.

We have also improved our benchmarking by adding new, more suitable performance criteria, testing of 4 and 5 cluster in silico mixtures, and thorough comparisons of SVclone to both PyClone and Battenberg.

Please find below our point by point response to the all reviewer comments.

Reviewer 1

1. Discussion of the coverage of data in PCAWG cohort is missing, as well as the overall impact of coverage on the performance of the tool.

Information on samples coming from the PCAWG cohort can be found in the main marker papers for the consortium (<https://doi.org/10.1101/162784>, <https://doi.org/10.1101/161562>). Our related manuscript on tumour heterogeneity (<https://doi.org/10.1101/312041>) presents an extensive analysis on the power to detect subclonal clusters/mutations, taking into consideration sequencing depth/coverage, tumour purity and copy number for SNVs. As we compute variant allele frequency in a similar way to SNVs, these analyses are equally applicable to SVs. We have more clearly articulated these challenges and referenced our related manuscript in the discussion (page 9). It is important to note that we now use the “number of reads per chromosome copy” (NRPCC) to determine which PCAWG samples have sufficient power to detect subclonal mutation clusters and only retain these for further analysis.

2. I am also sceptical about parameter setting in BIC and their effect of tool performance on data of different sequencing characteristics.

This is no longer applicable in our new model. We now run our clustering multiple times and select the solution with the best evidence lower bound (ELBO), see Supplementary Material Section 1.7 for details. With regard to different sequencing characteristics, please see point 1 above, where we articulate how we have addressed the impact of general data characteristics, such as sequencing depth, tumour purity and copy-number, on the tool.

3. The last [points], together with the tool's performance in the cases where tumour is composed of >3 clusters and considering different read depths, should be more thoroughly analyzed via simulated data.

Pure simulation of structural variants is extremely difficult and requires a large number of assumptions which generates potentially artificial data that does not reflect real world sequence characteristics. For this reason, we explicitly designed a performance assessment strategy which did not require purely simulated data. Instead, we sampled and mixed reads from previously sequenced human tumours to generate data with known subclonal structure while retaining real world sequence characteristics. These data have a fixed sequencing depth, unfortunately preventing us from exploring the effects of sequencing depth on performance. Despite this limitation, the simulated subclonal mixes generated from two prostate cancer metastases, rather than purely simulated data, better represents the noise present in real tumour sequence data and the challenge in identifying separable clusters of subclonal mutations. We believe assessing performance in the presence of real noise is preferable to assessing performance relative to depth. To offset this limitation we rely on the extensive discussion found in our manuscript (<https://doi.org/10.1101/312041>) on the effects of sequencing depth, tumour purity and copy number appear.

Furthermore, to better express the difficulty in identifying subclonal clusters of mutations in the presence of real noise, we have updated our performance assessment to highlight both the “ground” truth, plus the “obtainable” truth (the best possible separation of mutations into clusters). Please see Figure 2 and page 5 for a description.

In our updated manuscript we have also created two further *in silico* mixtures of real prostate metastases composed of 4 and 5 clusters each (a description of how clusters were generated and new performance results can be found in Figure 2, text on pages 13-14).

4. Sentence "The mutations belonging to each clone in a tumour can be interrogated using bulk whole-genome sequencing, with mutation detection subject to sequencing depth, tumour cellularity, clonality and mutation copy-number" should be made less explicit as there are other factors affecting mutation calling (for example, sequencing quality).

We have reworded this sentence to clarify that clonal analysis is subject to a greater number of factors (see page 2).

5. Description in Figure 1B is unclear. What do different colors (red, green, blue, grey) represent? The meaning of the last column in this figure ("adjust normal reads" etc.) isn't it obvious.

Figure 1b has been streamlined. Reads are now displayed all in blue, with red representing soft-clipped reads (which has been added to the figure legend). 'Adjust normal reads' has been changed to 'downscale normal read counts' to clarify meaning.

6. While presenting the error in inferring clonal frequencies in results section, it would be more convenient for the reader to interpret results if mean absolute error (MAE) is presented rather than mean squared error difference.

We thank the reviewer for this point which prompted us to rethink which metrics we use to assess performance. As our method is chiefly designed to determine the clonality of single tumour samples, we chose a number of metrics which reflect the ability of SVclone to provide useful information for downstream analysis in such a setting (motivated by the analyses appearing in our related paper (<https://doi.org/10.1101/312041>), among others). Our new benchmark metrics include: cluster number error, mean cluster CCF error, mean variant CCF error, mean multiplicity error and subclonal classification sensitivity and specificity (whether a variant is subclonal or not). A description of these metrics can be found on pages 6-7 and results are presented in Figures 2 and 3.

7. In the sentence "Both SVclone and PyClone had little difficulty in identifying minor subclonal clusters in the mixtures due to the lack of overlapping CCFs from clonal or major subclonal clusters." the second part of the sentence ("due to the lack ...") is not very clear. Could you please clarify.

The revised paper no longer contains this sentence.

8. How does the performance of SVclone compare to Pyclone in terms of number of inferred clusters and MAE if it is run in coclustering mode (i.e. using both SVs and SNVs) on in-silico mixtures of real tumour data presented in Figure 1D and Figure 1E?

As our new model clusters both breakpoint ends of each structural variant, it is no longer compatible with co-clustering SNVs. However, the method can be used to cluster SNVs independently, then the results can be combined to infer a single clustering result. SNVs and SVs can be post-assigned to clusters derived from this joint model. We have reported results from this approach in the updated manuscript which shows a slight performance improvement when tested on our *in silico* mixes (Supplementary Figure 7).

9. What tumour purity values were supplied to SVclone and (if any) Pyclone while analyzing abovementioned in-silico mixtures?

Both SVclone and PyClone utilised purity estimates obtained from Battenberg. We have clarified this in the methods section, page 14, and in Supplementary Information Section 3.

10. In the model selection part in the supplementary, the authors state "In selecting the best run, we also do not consider any runs where there is a single non-clonal cluster". What is the fraction of times (on, say, 100 different runs of the tool) where SVclone reports only 1 subclonal cluster in the analysis of *in-silico* mixture? Was this one of the reasons that tool correctly identified number of clusters? Note that this is also correlated to the above critics of testing the tool and its performance in the presence of more than 2 subclonal clusters since at this point it is unclear whether SVclone is under-clustering.

Our revised clustering model no longer discards runs with single non-clonal clusters. Using the updated model, we observed no instances of a single non-clonal cluster in the *in silico* mixtures. To estimate the proportion of times where only a single non-clonal cluster may be called across a representative set of samples, we considered the PCAWG results. We have added the following to the PCWAG methods section (page 14) to address this point: "We also tested the rate at which SVclone called single non-clonal clusters in PCAWG samples. Using a cutoff of < 0.7 cluster CCF, the SV clustering reported 9 samples with single non-clonal clusters across 1,220 PCAWG samples with (0.74%) with at least 10 SVs, and the SNV clustering reported 4 samples across 1637 (0.24%) samples with >10 SNVs. These results indicated that rates of under-clustering were low, and were similar across the SV and SNV clustering models."

11. In "with >10 SVs and SNVs" please make it very clear whether "SNV + SV > 10 " or "SNV >10 and SV >10 "

We have clarified that this refers to SNV >10 and SV >10 in the revised text, page 7.

12. In Figures 2E and 2F why there is no clear "border" between the right end of subclonal CCFs (Figure 2E) and left end of CCFs of clonal SV events (Figure 2F)? Is this a consequence of applied methodology or something else?

In the original version of the paper our analysis was performed on clusters of SVs rather than individual SVs. In the revised version, we use a structural variant-level CCF cutoff, informed by the optimal sensitivity and specificity derived from the *in silico* mixtures. A clear cutoff between the clonal and subclonal CCFs can now be observed in e and f of the revised Figure 5.

13. Recently, a tool for inferring trees of tumour evolution by the use of SVs was made available and accompanying method description presented in the manuscript that can be found at <https://www.biorxiv.org/content/early/2018/01/30/257014>. While I am aware that this is still not the peer-reviewed publication, considering the lack of methods for the particular problem, I would appreciate if the authors can compare against this method or justify if they are unable or unwilling to compare for whatever reason(s).

We would like to thank the reviewer for drawing our attention to this work. This method, now published in *Bioinformatics*, addresses an important aspect of estimating the clonality of tumour samples using SVs that is wholly complementary to SVclone's. The TUSV method relies on

mixtures of copy number states, presence/absence of breakpoints, and phylogenetic constraints to estimate tumour clonality. Whereas, SVclone is based on cancer cell fraction estimation of individual SVs. Comparing the performance of these algorithms would be limited to the number of clonal populations and their frequency. As the algorithm requires that the SVs be called using Weaver, the significant work required in performing this comparison outweighs the benefit. Therefore, we have elected not to compare to this method. Rather, we have mentioned this work in our discussion and suggested that both approaches could be run on a tumour sample to provide the most comprehensive view of SV clonality in a sample (page 9).

Reviewer 2

14. Notably, the approach underlying the method itself is fundamentally not novel and includes repurposing of a previously published statistical model, albeit on SVs (which is the new part).

In this re-submission we have fundamentally overhauled our clustering methodology by extending a variational Bayes approach we recently developed (<https://doi.org/10.1101/484402>) to specifically work with SV data. This novel approach takes advantage of one of the key differences between SNVs and SVs in that SVs have two breakpoint locations, with two VAFs, rather than a single VAF for SNVs. As the reviewer points out, we previously adapted a SNV clustering method by choosing only one of the two breakpoints. In our revised manuscript, our new method uses VAFs from both breakpoints simultaneously in the clustering, thus providing a novel methodology specifically designed for SVs. Furthermore, we explicitly handle subclonal background copy-number variants for both SNVs and SVs. This is particularly important for SVs so as to maintain as many variants as possible for clustering (see the revised Figure 3). This new approach, in combination with our VAF counting methodology for SVs, is fundamentally novel, and no longer relies heavily on PyClone's statistical model.

15. the association of sub-clonal balanced rearrangements with survival is only superficially treated, lacks validation and lacks appropriate assessment of possible confounding co-variates

In our revised manuscript we have taken into consideration further covariates in our survival analysis including stratification by age, number of SVs and histology. Where sufficient data was available, we have also considered association of the SCNR genotype with other genotypes (see point 18). We also demonstrate that SCNR samples are found across numerous cancer types (Supplementary Figure 6). In considering further validation, the unique scale and depth of the data generated from the PCAWG project has allowed us to consider survival of a previously unconsidered trait (prevalence of subclonal balanced rearrangements) and its relationship with patient survival. Similar data sets of this magnitude are not available making orthogonal validation difficult. Given our focus on the methods component of the manuscript, we consider any further biological validation out of scope for this work.

16. Proper benchmarking needs to be undertaken, comparing to other methods in the field. In particular THetA2, TITAN, Battenberg (compared to by authors in supplement), ReMixT and CloneHD are available methods that represent state of the art of copy number CCF.

In response to this request we began running some of the algorithms outlined above on our benchmark data. However, using these approaches “out of the box” did not necessarily reflect the optimal performance of each method. Indeed, our experience running multiple methods as part of PCAWG showed that when the methods were run by “experts” (the authors of each method) performance was much better than simply running the method with default parameters. In light of this observation, we believe a separate, more extensive benchmarking manuscript is required, rather than performing a wide benchmarking in this manuscript. Indeed, a number of benchmarking efforts are underway in this area (<https://doi.org/10.1101/310425>, <https://doi.org/10.1101/418780>), including efforts to get authors to run their own methods (<https://www.synapse.org/#!Synapse:syn2813581>). Rather than include more comparison methods in the revised manuscript, we have opted to explore in more depth the benchmarking against a Pyclone and Battenberg, extending the number of performance metrics and the number of clusters in our *in silico* mixture ground truth dataset (see Figure 3 and 4, pages 5-7).

17. The statement ‘One major advantage of using SVclone is that it can also be used in co-clustering mode, where CCF estimates can be given for SVs and SNVs simultaneously’. This is not substantiated with any data or analysis. The authors should show quantitatively the benefit of co-clustering in order to include such a statement, perhaps through exposing the increase in accuracy as a function of incremental inclusion of SNVs.

As referenced in the response to Reviewer 1’s comment (point 8), we no longer co-cluster SVs and SNVs, opting to cluster both separately. We do provide an option for combining these data by assigning SVs to SNV clusters. A quantitative comparison of the performance of this approach can be found in Supplementary Figure 7).

18. The authors present a finding related to breast, liver and ovarian cancer. All three tumor types have had recently described genome-based subtypes. The authors must account for these covariates in their analysis. It is well-established that the poor-outcome basal subtype of breast cancer harbours increased genomic instability. The Nik-Zainal paper describing 560 breast cancers ([doi:10.1038/nature17676](https://doi.org/10.1038/nature17676)) has accompanying hormone receptor status so at the very least, the authors should be able to determine if their finding is independent of ER negativity, basal gene expression subtype and/or global metrics of genome instability. For ovarian cancers the Wang et al study ([doi:10.1038/ng.3849](https://doi.org/10.1038/ng.3849)) shows an enrichment of fold-back inversions in a poor prognostic group. They validate on the ICGC dataset and identify cases with fold-back inversion enrichment. Is the finding of sub-clonal balanced rearrangements independent of this?

To account for as many covariates as possible, we stratified the survival analysis on detailed (tier 4) tumour histology, age at diagnosis and number of SVs. In the PCAWG data, breast cancers comprise only 10 of the 171 SCNR samples (5.8%), making stratifying on ER status difficult, and suggesting that ER status is unlikely to play a critical role to the observed genotype. To aid in the interpretation of the genotype across the PCAWG data, we have added Supplementary Figure 5 to illustrate SCNR samples across the 38 cancer types, where breast cancer can be seen to be ranked 4th in absolute number. Ovarian cancers were most prevalent in terms of absolute number of SCNR-positive samples. To address the point of fold-back inversion enrichment, we quantified amplification associated fold back inversions as previously described (<https://www.nature.com/articles/s41588-018-0179-8>) and performed an association analysis which showed no correlation between FBI rates and the SCNR phenotype (page 16 and Supplementary Figure 6).

19. What is the relationship of #s of SVs to the tumor content of the sample? How does this impact the ascertainment of subclonal CCF? How does tumor content influence the outcome associations as a co-variate?

We have added Supplementary Figure 4 to address this point. As can be seen in the figure, the SV number and purity show no correlation and has an R^2 coefficient of 0.0001. See response to point 1, where we address the remaining points.

20. What is the impact of sequencing depth on sensitivity to low CCF SVs? This and the preceding question are essential analyses to be presented to the readers.

See response to point 1.

21. The statement 'clonality of balanced genome rearrangements reveals functionally important and clinically relevant observations.' is not corroborated with data. The authors fail to identify any functional data or experiments to support this statement.

See response to point 15.

22. The statement 'Importantly, considering only the clonality of SNVs and/or SCNAs would have failed to reveal this information.' is unsupported by any analysis. It is entirely possible that SCNAs at similar CCF represent clones that are the latent cause for poor outcome. SCNRs might simply be part of the genomic landscape of the clone that has a phenotype. The authors would have to demonstrate the the SCNRs are actually independent of SCNAs at similar CCF to substantiate this claim.

We agree with the reviewer that strictly, this statement would require us to demonstrate that SCNAs could not be used as a proxy to identify clones with the SCNR phenotype. As our

analysis is not done at a clone level, rather, it is performed based on whether an event is clonal or subclonal, this analysis is not possible. Therefore, we have removed this statement.

23. The authors somehow seem to miss an opportunity to associate their findings with driver mutations in these genomes as a route to indicating mechanism for SCNRs. Are there significant associations to the prevalence of driver mutations in the cases harboring SCNRs?

We have now performed an analysis on the potential functional effects on driver mutations by the SCNRs (page 8).

24. The method itself relies heavily on the PyClone probabilistic model (doi:10.1038/nmeth.2883), repurposed for clustering SVs. This is not properly acknowledged in the main text anywhere and represents a major oversight/omission. (I note it is referenced in the supplementary info). Also the graphical model (very similar to PyClone) is never referenced in the main, nor supplementary text. If the authors are claiming novelty, they should establish how their method differs from the PyClone approach.

SVclone no longer relies on the PyClone probabilistic model.

25. The authors rely on a linear scaling of the SV supporting reads dependent on the SV type. The authors test on a prostate sample which are known to harbour primarily blunt end joined DNA breaks. Have the authors tested the robustness of their scaling method when either microhomology or non-templated inserted sequence is present at the breakpoint? What is the effect of read length?

We thank the reviewer for this insightful comment which prompted us to look more closely at breaks with microhomology. We identified that due to the imprecise genomic location of breaks with microhomology our method double counted a small number of normal reads, which ended up being the cause of the systematic bias observed in the simulated VAF data. We have now implemented a procedure which accurately estimates the VAF for breaks with microhomology up to 6bp. As such, we no longer require the linear scaling of all events and this has been removed from the manuscript. We have also added the following to the Supplementary Information Section 1.4.1: "In order to determine whether micro-homology was likely to play a large role in the read counting process, we analysed the distribution of breaks containing micro-homologies across the PCAWG samples used in the paper analysis (using PCAWG's consensus SVs v1.6). We found that the mean and median micro-homology lengths were 1 and 2.4 respectively. Micro-homologies \leq 6bp in length are handled by the variable threshold used by our read counting step. 6.17% of SVs had micro-homologies greater than 6bp and <1% of SVs had micro-homologies greater than 20bp. Although small in number, these events may have noisy VAF estimates and such it is recommended they be filtered out."

26. Do the authors observe multi-modality in the posterior distributions of CCF for individual events? This is likely going to be the case in many situations where the relative combination of ploidy, tumor content and CCF is unidentifiable? How, in turn does this impact the distribution over clusterings of the events. The authors should show co-clustering distributions computed over the 25,000 MCMC samples.

The CCF posterior is naturally multi-modal even if there is no uncertainty relating copy number and tumour content. It arises from any Bayesian treatment of mixture models, as cluster labels are interchangeable in the marginal likelihood of the model, i.e. label switching problem. Our updated variational inference method is less prone to this problem as it optimises a fixed-form distribution. In our case, it is Gaussian, hence no multi-modality. We agree that the identifiability of CCF is low. In our model, CCF and multiplicity are both unknown. They participate in the model in a co-linear relationship. Unfortunately, in theory, there could be an infinite number of solutions. Our model addresses this fundamental lack of information by assuming many variants share the same CCF (clustering), and aim for a conditional posterior (Gaussian) of CCFs given point estimates of multiplicities. The identifiability is further strengthened by only searching within sets of finite possible multiplicities. To explore this empirically, we have considered the “mean multiplicity error” as one of the standard six metrics we have used to evaluate the clustering performance. Mean multiplicity error measures optimal multiplicity assignment (given true cluster means) minus the estimated multiplicity from inferred cluster means. As can be seen in Figure 4, mean multiplicity error of SVs is similar to multiplicity error in SNVs, indicating that our clustering methodology performs comparably in minimising error from multi-modal distributions.

27. ‘Clusters which have no variants assigned are discarded’. How is it possible in a DP to have an empty cluster?

To the best of our knowledge, empty clusters arise in several inference methods for DP and its variants. Radford Neal’s classic review on MCMCs for DP suggests removing empty clusters at every iteration to improve the mixing properties of the samples (DOI: 10.2307/1390653). In the finite mixture (truncated DP) model employed in the previous version of the manuscript, empty clusters arose from the Dirichlet distribution over mixing weights. The Dirichlet distribution is known to have a shrinkage effect when its concentration parameter is small. Our variational inference in the updated manuscript also removes empty clusters in its QC steps. The posterior inference model for determining cluster assignments is described in Supplementary Information Section 1.7.

28. There is a comparison to Battenberg CCFs presented only in the supplementary information. In Supplementary Fig 7, the authors present correlations. The data appear substantially uncorrelated across all size categories. This raises a serious concern to this reviewer and suggests quite unexpected behaviour from SVClone. Which method is correct? Why? Under what circumstances should a user trust SVClone vs Battenberg (or for that matter, TITAN, THetA2, ReMixT once those comparisons have been made)?

This needs appropriate treatment using established comparison methods fitting of a computational methods paper.

We agree that the initial comparison between Battenberg and SVclone did not show *strong* correlation, however, the comparison did yield significant correlation coefficients ranging from 0.28 to 0.54. In our experience we would not expect *strong* correlation between individual events. Therefore, in the revised manuscript we have updated our comparison, moving away from correlation between individual events to metrics which look at multiple events: mean variant CCF error, subclonal classification specificity, and sensitivity. As we have noted in point 16, SVclone's variant CCF estimation performs similarly to Battenberg, with Battenberg performing better in low-cluster mixtures (3 clusters), and SVclone performing better in high-cluster mixtures (4 and 5 clusters) page 6. This makes sense as Battenberg is restricted to a 2-clone model.

29. The authors justification of the binomial distribution is convoluted and unconvincing. First the authors remove VAF outliers before testing for goodness of fit which surely improves the goodness of fit to a distribution with constrained variance such as the Binomial. The authors use a chi squared test which is only appropriate for categorical data, confusingly with two Poisson distributions instead of a Binomial. The authors should fit the data to both a Binomial and Beta Binomial (the distribution used by PyClone) or other distribution and use a likelihood ratio test.

We have reworked the justification for the binomial distribution (described in Section 2.2 of the Supplementary Information). We now use a likelihood ratio test on the clonal SVs using the patient metastasis samples we used for the in silico mixtures. At these coverage and purity levels (common for WGS data), we found that 89% of SVs were consistent with the binomial distribution, indicating that the distribution is an appropriate choice at the moderate purity and coverage levels. In addition, our variational formulation produces a similar effect to a Beta-Binomial model as the assignment probability is computed as an expectation of the Binomial distribution with respect to the posterior CCF distribution. Therefore, the uncertainty within the probability of success is integrated out when making assignments. The difference is that the uncertainty is a Gaussian in our model, and the uncertainty is Beta-distributed in the Beta-Binomial distribution. The benefit of our choice is a fully tractable variational approximation in which all its parameters can be efficiently estimated. Whereas in the Beta-Binomial case, its key overdispersion parameter is difficult to estimate with 50x depth. The difficulty is evident in the high variance and lack of clear convergence observed in PyClone's MCMC traces of its overdispersion parameter (Supplementary Figure 8).

30. The modified BIC has no theoretical justification. The main motivation for using a DP is to obviate model selection.

As referenced in point 2, we no longer use the BIC for model selection, instead we use the best evidence lower bound which is a direct approximation to the ideal model selection criteria, marginal likelihood.

31. Overall the paper is quite short and contains a lot of relevant material in the supplementary info that would be of interest to the reader if presented in the main text.

We have substantially rewritten the paper and moved much of the Supplementary material to the main text.

32. There are supplementary figures not referenced in the main text.

We have ensured that all Supplementary Figures are now referenced in the main text.

Reviewers' comments:

Reviewer #1 (Remarks to the Author):

The authors have made considerable changes in the method and added additional analysis. Some of my previous criticism is alleviated by changes in the method. SVclone has been used as one of the important tools in PCAWG project for analysis of heterogeneity with results available in 2 bioRxiv preprints.

In addition to difference in the methodology used, one of the important claims of novelty against SNV based methods (e.g. PyClone), which use read counts from only a single genomic position, is the consideration of read counts at the ends of an SV (i.e. multiple pairs of (reference, variant) read counts), under the assumption that they share CCF, but have possible difference in copy number. While this is in principle methodologically novel in comparison to PyClone and other SNV-clustering based methods, what is the evidence that in the practical applications this will yield some advantage to SVclone? For what percentage of SVs it is important to consider endpoints separately (i.e. what percentage has different copy numbers at the ends)? Also, such SVs are likely residing in regions having increased genomic instability and complex patterns of overlapping copy number changes are not unlikely. Such CNAs imply the existence of multiple (healthy + multiple cancerous) populations of cells with respect to copy number state of these regions and estimating these numbers is very hard (even detecting and estimating allelic copy numbers of non-clonal CNAs is known to be very arduous task). SVclone relies on estimates of copy numbers that are in many cases inaccurate or incomplete. How inaccurate copy-number estimates and incomplete picture of overlapping CNA events (e.g. one occurring before SV event, one after) affect the performance of SVclone? In other words, I agree that considering the endpoints separately differs in terms of the methodology in comparison to the previously published methods, but am currently sceptical about the practical advantage and benefit of this over the simpler approach.

I also did not understand why comparison against TUSV is omitted? In addition to the comparisons or arguments why it is impossible to compare the two methods, some more comprehensive discussion of the difference between SVclone and TUSV is required, especially considering the fact that TUSV was published before re-submission of SVclone and presented at ISMB conference last year.

Reviewer #3 (Remarks to the Author): Replacement for Reviewer #2

Cmero and co-authors present a new tool for inferring cancer cell fraction of structural variants. Unlike previous approaches that use SNV and copy-number information, SVclone relies on quantification of reads that overlap SV breakpoints and 'unaffected' regions to quantify CCF. This allows to extend the scope of the CCF calculation embrace both – copy number and copy neutral structural variants. The quantification and modeling approaches taken in this paper is new and sound.

Cmero et al performed in silico mixing simulation using prostate cancer datasets. I consider this type of benchmarking to be better than simulation of reads, as the former uses real data with characteristic noise profile that is difficult to represent in read-base simulations. The authors compare the performance of their tool with approach using SNV information using five different metrics and found them to be comparable (which is good news given the smaller number of SVs compared to SNVs).

Further paper illustrates the utility of the new analysis approach by analyzing PCAWG dataset of

1705 tumor genomes. They find interesting observation that subclonal copy neutral rearrangements might be linked to survival of cancer patients. As the ability to analyse copy-neutral SVs is a characteristic feature that distinguishes SVclone from other tools, this a very good illustration of application of their tool to real world problem. This is clearly an advantage for a paper primarily aimed at describing a new method for CCF inference.

It is evident that paper was substantially reworked during revision and most of the points raised earlier were addressed. Overall, I find this to be interesting paper that describes a new tool, that will be very useful for cancer genomics community.

Dear Reviewers,

We would like to thank the reviewers for taking the time and effort to reconsider the manuscript and the associated changes.

We have carried out a series of performance assessment analyses which address the reviewer concerns:

1. As suggested we have compared our new dual-end SV clustering model to a standard single-end model (as employed when clustering SNVs).
2. To further demonstrate the advantages of the dual-end model, as well as consider the effects of inaccurate copy-number estimation on SVclone's results, we have also performed a series of copy-number perturbation experiments, modifying the major allele copy-number, as well as subclonal copy-number fractions.
3. Additionally, we demonstrate that the dual-end model is more robust to copy-number perturbation than the single-end model.
4. Lastly, we have considered the TUSV method and have provided a more detailed discussion.

Please find our point-by-point analysis below:

Reviewers' comments:

Reviewer #1 (Remarks to the Author):

1. In addition to differences in the methodology used, one of the important claims of novelty against SNV based methods (e.g. PyClone), which uses read counts from only a single genomic position, is the consideration of read counts at the ends of an SV (i.e. multiple pairs of (reference, variant) read counts), under the assumption that they share CCF, but have possible difference in copy number. While this is in principle methodologically novel in comparison to PyClone and other SNV-clustering based methods, what is the evidence that in the practical applications this will yield some advantage to SVclone?

To address this point, we compared SVclone's dual end model to (SVclone's) single-end model (a similar statistical model to PyClone), using our 3-cluster *in silico* mixing dataset. To reduce the SV data to single-end, we ran both ends of each mixture separately through the SVclone's SNV model, which only considers one background copy-number state and one normal read count. The revised results section discusses the results:

"Figure 4 shows that dual-end model outperforms the single end model across mean variant CCF error, mean multiplicity error, and mean cluster CCF error across almost all mixes. Only the cluster number of the 50-50 mix was incorrectly inferred, compared to the single-end model which was correct. However, we would expect only two clusters given the 50-50 mixture split

and thus the dual-end model's result is likely more parsimonious with the data. Interestingly, the single-end model showed a higher subclonal classification sensitivity, but a lower specificity than the dual-end model. Given that this metric represents a trade-off of sensitivity and specificity, we generated an ROC curve (Supplementary Figure 10). Considering the AUC indicates that the dual-end model is preferable (AUC of 0.8234 versus 0.8095 for the dual and single-end models respectively)."

2. For what percentage of SVs it is important to consider endpoints separately (i.e. what percentage has different copy numbers at the ends)?

We have quantified the percent of SVs with heterogeneous background copy-numbers in the PCAWG (Supplementary Table 4 in the revised manuscript). We observe a range of background copy-number heterogeneity across the cohort, with a minimum of 34% in non-Hodgkin lymphoma and a maximum of 74% in colorectal adenoma. The median across all cancer types was 53%.

3. Also, such [*background copy-number heterogenous*] SVs are likely residing in regions having increased genomic instability and complex patterns of overlapping copy number changes are not unlikely. Such CNAs imply the existence of multiple (healthy + multiple cancerous) populations of cells with respect to copy number state of these regions and estimating these numbers is very hard (even detecting and estimating allelic copy numbers of non-clonal CNAs is known to be very arduous task). SVclone relies on estimates of copy numbers that are in many cases inaccurate or incomplete. How inaccurate copy-number estimates and incomplete picture of overlapping CNA events (e.g. one occurring before SV event, one after) affect the performance of SVclone? In other words, I agree that considering the endpoints separately differs in terms of the methodology in comparison to the previously published methods, but am currently sceptical about the practical advantage and benefit of this over the simpler approach.

To address this point, we performed a series of experiments where we introduced copy-number noise into the 3-cluster *in silico* mixtures. We selected the 70-30 mixture to perform these experiments, given that it had the lowest variant CCF error. The 001bM and 001gM samples used in the *in silico* mixtures have 30% and 37% background SCNA heterogeneity (copy-number states are different between an SV's ends) respectively, while the mixtures have 24-27% heterogeneity (Supplementary Table 3), indicating there may be some 'averaging' of copy-number states occurring due to the mixing. In order to correct for this, given the PCAWG copy-number heterogeneity statistics (see response to point 2), we randomly discarded SVs until 50% of total SVs had heterogenous background copy-number states. This resulted in 90 total SVs. The experiment is outlined in the results of the revised paper:

"[We] perturbed copy-number in the following ways: i) major allele copy-number - 1, ii) major allele copy-number + 1, and iii) subclonal copy-number fraction + 0.3, where, if the resulting

copy-number fraction would be >0.9 , we instead subtract 0.3 (see methods for further details). We performed these experiments for the dual-end model, perturbing one side and both sides in separate runs. As expected, we found that the CN-perturbed runs showed slightly worse performance across the measured metrics compared to the unperturbed runs (Supplementary Figure 11). In general, variant-level metrics were more severely affected than cluster-level metrics. All perturbations performed similarly, with CN - 1 (experiment i) on both sides being the most affected scenario. Mean variant CCF was most significantly affected with a 0.27 error in the CN - 1 scenario on both sides (compared with 0.07 in the unperturbed model). Mean cluster CCF error was only mildly affected, but was also most significant for the CN - 1 on both sides scenario (0.16 versus 0.11 ME in the unperturbed data). The CN - 1 experiments were the only ones that caused an error in the cluster number. Supplementary Figure 12 shows the effects of the perturbation experiments on the single-end model versus the dual-end model (where only one side is perturbed). The dual-end model was more robust to perturbation across all metrics for all perturbations except for cluster number with the CN - 1 experiment (where one extra cluster was called), subclonal classification sensitivity in the CN - 1 experiment and a slightly worse mean multiplicity error in the CN + 1 scenario. Interestingly, mean cluster CCF error was still lower in the over-clustered case. Importantly, the mean variant CCF error and mean cluster CCF error were lower in all cases when considering the perturbed dual-end model versus the perturbed single-end. In summary, these data show that the dual end model is more robust to copy number noise than the single end. Copy number addition errors were better tolerated than subtraction errors, and a mis-estimation of copy-number fraction resulted in errors somewhere between the two. However, mean cluster CCF error and cluster number were minimally affected, suggesting that poor CN estimation effects are largely restricted to errors in variant-level estimates.”

4. Why [is] comparison against TUSV is omitted? In addition to the comparisons or arguments why it is impossible to compare the two methods, some more comprehensive discussion of the difference between SVclone and TUSV is required, especially considering the fact that TUSV was published before re-submission of SVclone and presented at ISMB conference last year.

We chose not to compare the performance of SVclone to TUSV in the manuscript for three reasons: 1) TUSV is primarily designed to work with multiple samples, whereas SVclone is designed to work with single samples; 2) There are serious technical hurdles that make a rigorous and fair comparison very time consuming; and 3) The algorithms tackle different aspects of the SV evolution inference problem and are therefore only comparable on a subset of outputs. Please find below more detailed explanations of each of these reasons and how we have modified the manuscript to clarify these points.

- 1) *Single sample versus multi-sample inference.* SVclone is designed to work with single tumour biopsy samples sequenced to a depth of approximately 50X. The goal of SVclone is to infer cancer cell fractions (CCF) of SVs, which are crucial summary statistics of the phylogenetic tree that can answer many important questions in cancer biology, for example which SVs are early evolving and which are late.

Building the full phylogenetic tree is a far more difficult and challenging problem. To attain reasonable performance, WGS data at sequencing depths in excess of 100X and multiple samples from the same patient are required. TUSV aims to tackle phylogenetic tree building using a matrix factorisation-based deconvolution model, which inherently performs best when it can draw information from multiple samples.

This clear difference in the goals of SVclone and TUSV also shows when comparing their runtimes. As the TUSV authors reported, when testing with 28 out of 59 WGS TCGA single-samples, TUSV ran more than 2 days or required more than 128 Gb of RAM. With the same amount of time and 16GB of RAM, SVclone can analyse > 2000 WGS single-samples from the PCAWG cohort. This demonstrates that SVclone is more suitable to single-sample WGS, which is a much more prevalent data type to date.

- 2) *Technical hurdles for fair method comparison.* The differences between SVclone and TUSV also show on a technical level: SVclone takes as input, SV calls, copy-number calls, and an associated bam file. In contrast, TUSV takes as input, SV clonal multiplicity mapped to copy number calls. The initial multiplicity estimates input into TUSV are calculated by an independent program called WEAVER. To perform a robust and fair comparison between the approaches it would be necessary to disentangle the multiplicity estimation of SVclone (which is a core part of the model and not an independent step), compare it to WEAVER, and then find a way to fix multiplicity between the algorithms to compare other outputs. This would be a significant undertaking and is best suited to a separate benchmarking manuscript.
- 3) *SVclone, TUSV, and other approaches address different aspects of tumour evolution.* We compared the capabilities of TUSV (and WEAVER) along with Fan et al.'s SV VAF calculation approach, and another recently published method in the area: Meltos (Ricketts 2019). This revealed that each of these methods, along with SVclone, tackle different subtasks of the problem of inferring the evolutionary history of SVs from WGS data.

The subtasks we identified were the inference of: variant allele frequencies of SV breakpoints, number of DNA copies harbouring SV breakpoints (also known as multiplicity), the cancer cell fraction of SVs, mixing proportions of clones/clusters, clone copy number and a clone phylogeny.

The following table shows how different methods tackle different subset of these challenges. In particular, TUSV is the only method that computes copy number profiles for individual clones, while SVclone is the only method that computes the CCF for each SV (although an estimate of CCF can be obtained by further processing of TUSVs output).

	Fan et al	WEAVER	TUSV	MELTOS	SVclone
VAF of SV	X	X		X	X
Clonal Multiplicity		X			X
Subclonal Multiplicity			X		X
CCF of SVs					X
Clone CCF			X		X
Clone CN			X		
Clone Phylogeny			X	X	

In light of these results we have added a section to the discussion (see below) highlighting these differences.

“Inferring the evolutionary history of SVs from whole-genome sequence data is a challenging problem. One of the key goals in the field is to derive a clone tree which depicts the acquisition of SVs over time and their relationship to clonal expansions during tumour evolution. To achieve this, a number of key variables must be inferred from the data: variant allele frequencies of SV breakpoints; number of DNA copies harbouring SV breakpoints (also known as multiplicity); the cancer cell fraction of SVs; cancer cell fraction of clones; and a clone phylogeny. No one method exists that can simultaneously infer all variables, but rather existing methods tackle subsets: Fan, et al.: VAF³¹; WEAVER: VAF + clonal multiplicity³²; TUSV: subclonal multiplicity + clone CCF + phylogeny + (additionally) clone copy number³³; Meltos: VAF + phylogeny³⁴; and SVclone: VAF + subclonal multiplicity + approximate clone CCF + SV CCF. At present these methods need to be combined to achieve a more complete picture of the evolution of SVs (e.g. WEAVER + TUSV³³ or SVclone + Meltos³⁴). Thus there remains an opportunity for future development of an algorithm which can simultaneously infer all variables.”

REVIEWERS' COMMENTS:

Reviewer #1 (Remarks to the Author):

My main concerns regarding comparisons to the available alternative have been addressed in the Response Letter.

I recommend that content added between revisions is colored.